# PARP3 is a promoter of chromosomal rearrangements and limits G4 DNA

Tovah A. Day[1], Jacob V. Layer[1], J. Patrick Cleary[1], Srijoy Guha[1], Kristen E. Stevenson[2], Trevor Tivey[1], Sunhee Kim[1], Anna C. Schinzel[3], Francesca Izzo[1,3], John Doench[3], David E. Root[3], William C. Hahn[1,3], Brendan D. Price[4] & David M. Weinstock[1,3]

Chromosomal rearrangements are essential events in the pathogenesis of both malignant and nonmalignant disorders, yet the factors affecting their formation are incompletely understood. Here we develop a zinc-finger nuclease translocation reporter and screen for factors that modulate rearrangements in human cells. We identify UBC9 and RAD50 as suppressors and 53BP1, DDB1 and poly(ADP)ribose polymerase 3 (PARP3) as promoters of chromosomal rearrangements across human cell types. We focus on PARP3 as it is dispensable for murine viability and has druggable catalytic activity. We find that PARP3 regulates G quadruplex (G4) DNA in response to DNA damage, which suppresses repair by nonhomologous end-joining and homologous recombination. Chemical stabilization of G4 DNA in $PARP3^{-/-}$ cells leads to widespread DNA double-strand breaks and synthetic lethality. We propose a model in which PARP3 suppresses G4 DNA and facilitates DNA repair by multiple pathways.

[1] Department of Medical Oncology, Dana-Farber Cancer Institute, 450 Brookline Avenue, Boston, Massachusetts 02215, USA. [2] Department of Biostatistics and Computational Biology, Dana-Farber Cancer Institute, 450 Brookline Avenue, Boston, Massachusetts 02215, USA. [3] Genetic Perturbation Platform, Broad Institute of MIT and Harvard University, 415 Main Street, Cambridge, Massachusetts 02142, USA. [4] Department of Radiation Oncology, Dana-Farber Cancer Institute, Harvard Medical School, 450 Brookline Avenue, Boston, Massachusetts 02215, USA. Correspondence and requests for materials should be addressed to T.A.D. (email: tovah_day@dfci.harvard.edu) or to D.M.W. (email: dweinstock@partners.org).

Chromosomal rearrangements, including translocations, are essential events in the pathogenesis of malignant and nonmalignant disorders[1–3]. Both germline defects in DNA repair and certain clastogens increase the risk for oncogenic rearrangements that drive tumour progression and therapeutic resistance[4,5].

Rearrangements are believed to occur when two DNA double-strand breaks (DSBs) fuse inappropriately. Because DSBs pose a serious threat to genomic stability, multiple mechanisms exist to repair them. Nonhomologous end-joining (NHEJ) pathways repair DSBs throughout the cell cycle by direct re-ligation of the broken ends; nucleotides may be lost or added at the broken ends before ligation, resulting in sequence modifications. NHEJ pathways include classical NHEJ (c-NHEJ), which is essential for ionizing radiation (IR) resistance and V(D)J recombination, as well as one or more pathways of alternative NHEJ. In contrast to NHEJ, homologous recombination (HR) repairs DSBs during S and G2 phases by copying a portion of the sister chromatid to bridge the lesion.

Because they largely result from the fusion of nonhomologous sequences, chromosomal rearrangements are thought to primarily occur through NHEJ. Indeed, a recent study demonstrated that loss of XRCC4-LIG4, the essential ligase complex involved in c-NHEJ, reduces translocation formation between endonuclease-induced DSBs in human cells[6].

Chromosomal rearrangements between endonuclease-induced DSBs occur in *Saccharomyces cerevisiae* at frequencies of up to 50% (ref. 7). These high rates have allowed for agnostic strategies to identify factors that modulate rearrangements[8–10]. In contrast, rearrangements in mammalian cells between endonuclease-induced DSBs are much less frequent, which has precluded an effort to screen for modulators. As a result, only a small number of c-NHEJ and alternative NHEJ factors are known to affect rearrangements[6,11–15]. We developed a tractable strategy for inducing and quantifying targeted translocations across a range of human cell types. Using this approach, we perform a medium-throughput loss-of-function screen to identify factors that promote or suppress chromosomal rearrangements in human cells.

## Results

### A screen for modulators of chromosomal rearrangements.
Previous approaches to induce targeted translocations in human cells have utilized PCR to quantify frequency[6], which precluded the use of pooled loss- or gain-of-function screens. To facilitate screening for genetic factors that modulate chromosomal rearrangements, we established a flow cytometry-based zinc-finger nuclease translocation reporter (ZITR) assay by randomly integrating either a *GFP* or *CD4* transgene downstream of sequence from the *AAVS1* locus (Fig. 1a; Supplementary Fig. 1a). Cleavage of the integrated *AAVS1* sequence by zinc-finger nucleases (ZFNs)[16] and cleavage at the endogenous *AAVS1* locus on chr.19 followed by rearrangement between the heterologous DSBs resulted in transgene expression in 0.5–1.5% of cells (Fig. 1b; Supplementary Fig. 1b,c).

We performed a short hairpin RNA (shRNA) screen targeting 169 DNA repair-associated genes with 5 shRNA/gene plus controls (Fig. 1b–d; Supplementary Data 1). A549 lung adenocarcinoma cells carrying a *GFP*- or *CD4* reporter were transduced with individual shRNAs at high multiplicity of infection and then pooled. After 7 days, ZFNs were delivered by adenovirus and transgene-positive cells were flow-sorted 48 h later. In total, we performed 7 replicates (3 GFP-reporter and 4 CD4-reporter) with $2.5 \times 10^7$ shRNA-expressing cells per replicate. We quantified the representation of individual shRNAs within transgene-positive and transgene-negative populations by

massively parallel sequencing and compared the abundance of each shRNA in these two populations.

Of the 169 genes, 5 merited further consideration, based on $\geq 2$ shRNAs with $\geq 2$ fold change in the same direction compared to controls, without causing significant toxicity (Fig. 1e; Supplementary Fig. 2a; Supplementary Data 2). It is likely that additional factors assayed in the screen are capable of affecting translocation frequency but did not meet our threshold for further consideration because of either incomplete gene suppression (for example, KU70; Supplementary Fig. 2b) or excessive toxicity from the shRNA (for example, MRE11 or RFC1; Supplementary Fig. 2c).

To validate the five hits in a separate lineage, we performed the ZITR assay in HeLa cervical adenocarcinoma cells and confirmed that all five hits affected rearrangement frequency as expected. Specifically, RAD50 and UBC9 suppressed rearrangements, while 53BP1, DDB1 and poly(ADP)ribose polymerase 3 (PARP3) promoted them (Fig. 2a–e). In validation, all 5 (100%) genes met the criteria of $\geq 3$ shRNAs resulting in the predicted phenotype (Fig. 2) without causing significant changes in cell cycle distribution (Supplementary Fig. 3). We observed striking correlations between the extent of knockdown and fold effect on rearrangement frequency, with $R^2 > 0.5$ for 3 of the 5 factors (Fig. 2). Thus, multiple factors from diverse protein families have dose-dependent effects on chromosomal rearrangements.

Among the five factors, *rad50* (refs 8–10) and *ubc9* (ref. 17) were previously identified as suppressors of translocations in yeast. 53BP1 is known to promote long-range end-joining events, including class switch recombination[18]. However, DDB1 and PARP3 have not been previously implicated as promoters of chromosomal rearrangements.

### PARP3 promotes rearrangements across lineages and reporters.
We elected to focus on PARP3, as it has druggable catalytic activity[19] and is dispensable for murine and cellular viability[20,21]. The PARP family of 17 proteins contains 3 enzymes (PARP1, PARP2 and PARP3) that have been implicated in DNA repair. PARP3 is known to cooperate in the recruitment of DNA repair factors, including mediators of c-NHEJ, and PARP3 loss delays the repair of DSBs[22,23]. A recent study catalogued PARylation targets of PARP1, PARP2 and PARP3; PARP3 had the highest number of total targets and only $\sim 10\%$ of these were shared with the other PARP enzymes[24], indicating unique but poorly elucidated cellular roles for PARP3.

Three different *PARP3* shRNAs reduced chromosomal rearrangement frequency between 60 and 70% in HeLa cells (Fig. 3a,b). We considered multiple ways in which this observation could be the result of a confounding effect from PARP3 knockdown. Importantly, PARP3 knockdown did not affect either: (1) the extent of cleavage at the targeted locus 18 h after ZFN transduction, (2) cell cycle distribution, (3) the number of basal DSBs (that is, without ZFN transduction), or (4) the ability of IR to induce cell cycle checkpoint activation (Fig. 3c; Supplementary Fig. 4a–c). PARP3 re-expression after depletion with an untranslated region-directed shRNA rescued the shRNA effect on rearrangements (Fig. 3d,e). Notably, overexpression of PARP3 did not further increase rearrangement frequency above the frequency in cells transduced with control shRNA (Fig. 3d,e). To test the effects from complete PARP3 loss, we generated $PARP3^{-/-}$ A549 cells (Supplementary Fig. 4d) carrying the *AAVS1* ZITR. As expected, re-expression of wild-type PARP3 in these cells increased the frequency of ZFN-induced rearrangements (Fig. 3f,g).

Next, we devised a CRISPR translocation reporter (CRITR) assay in unmodified 293T human embryonic kidney cells by targeting CAS9-mediated DSBs downstream of the constitutively

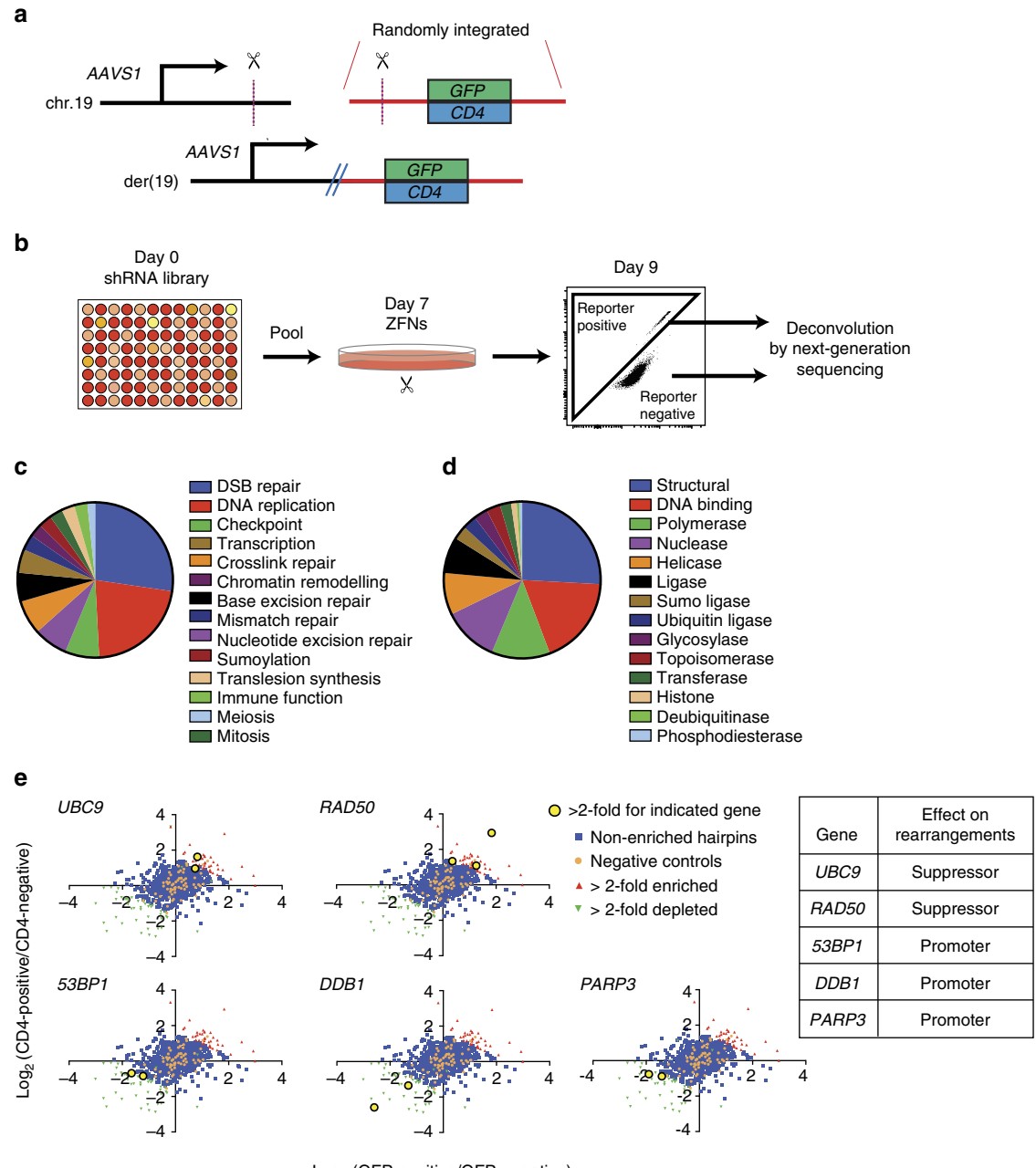

**Figure 1 | Identification of promoters and suppressors of chromosomal rearrangements in human cells.** (**a**) Flow cytometry-based assay for chromosomal rearrangements. Scissors, *AAVS1* zinc-finger nucleases (ZFNs). Dotted lines, targeted cutting at *AAVS1* recognition sequence upstream of either *GFP* or *CD4*. (**b**) Design of shRNA rearrangement screen. A549 cells harbouring a randomly integrated *GFP* or *CD4* reporter were transduced clonally with 1 of 966 shRNA, pooled and $2.5 \times 10^7$ shRNA-expressing cells were infected with *AAVS1* ZFN adenovirus, and flow-sorted by transgene expression. (**c**) Representation of different pathways in the shRNA library of 169 genes. (**d**) Representation of different protein functions in the shRNA library of 169 genes. (**e**) Gene hits identified in the shRNA screen. Average of $\log_2$ reads in transgene-positive population/average $\log_2$ reads in transgene-negative population for GFP replicates ($x$ axis) and CD4 replicates ($y$ axis).

expressed *CD71* promoter on chr.3 and upstream of the non-expressed *CD4*-coding sequence on chr.12. Translocation between these DSBs results in aberrant CD4 expression on der(3) (Fig. 3h,i). *PARP3*$^{-/-}$ cells had significantly reduced translocation frequency compared to wild-type cells (Fig. 3k,l). As with the ZFNs, loss of PARP3 did not affect the extent of cleavage by CAS9 at the target site (Fig. 3m). *PARP3* knockdown had a similar effect on translocations to *PARP3* deletion (Supplementary Fig. 4e,f). We also generated 293T cells lacking either *53BP1* or *LIG4*. These cells had a similar decrease in translocation

frequency using the CRITR as *PARP3*$^{-/-}$ cells (Fig. 3k–o); the reduced frequency of translocations was in *LIG4*$^{-/-}$ cells was previously shown using a PCR-based assay[6].

PARP3 is known to localize to laser-induced DNA damage[20]. To confirm that PARP3 localizes to a single, targeted DSB, we used CAS9-based transgenesis to knock-in a haemagglutinin (HA) tag at the endogenous *PARP3* locus. Chromatin immuno-precipitation (ChIP) using an anti-HA antibody confirmed that PARP3 accumulates at sequences flanking a CAS9-mediated DSB (Fig. 3p; Supplementary Fig. 4g).

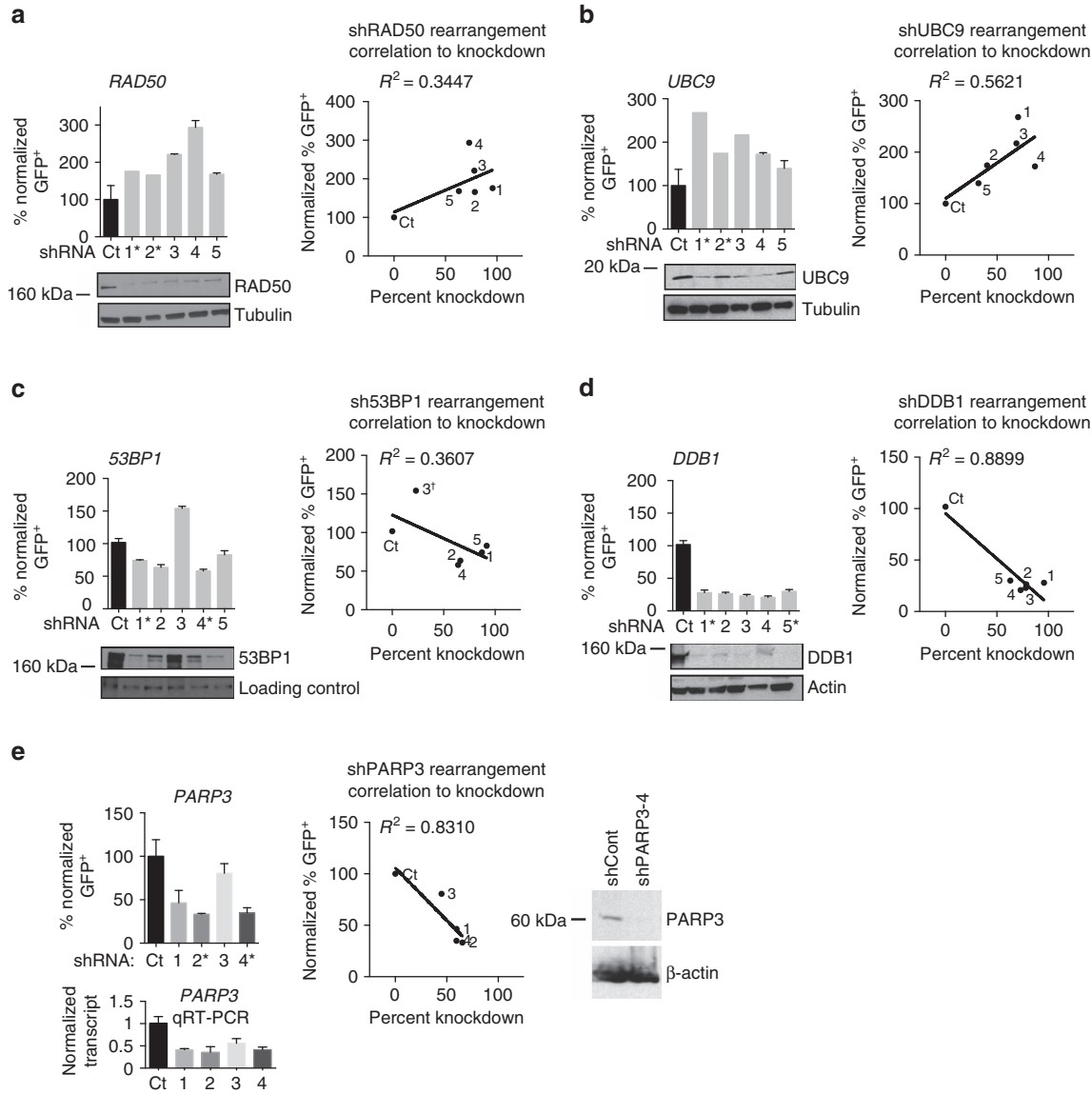

**Figure 2 | Validation of factors that modulate chromosomal rearrangements in human cells.** Normalized frequency of rearrangements, immunoblots and correlation of rearrangement frequency with the degree of knockdown in HeLa ZITR cells with shRNA-mediated knockdown of RAD50 (**a**), UBC9 (**b**), 53BP1 (**c**), DDB1 (**d**) and PARP3 (**e**). Data in bar graphs are presented as mean ± s.e. of $n = 3$. Immunoblots were quantified using ImageJ software and normalized to β-actin, tubulin or loading control. The dagger in **c** indicates that this data point (shRNA3) was excluded from the $R^2$ calculation as it resulted in 53BP1 overexpression. The shRNAs that scored in the screen are marked by asterisks. To compare rearrangement frequency with the degree of knockdown, we performed linear regression ($R^2$). $P$ values for $R^2$ calculations are as follows: RAD50, $P = 0.2206$; UBC9, $P = 0.0861$; 53BP1, $P = 0.2841$; DDB1, $P = 0.0048$; PARP3, $P = 0.0311$.

**PARP3 loss sensitizes cells to pyridostatin**. To further explore the mechanism of PARP3 in mediating DSB repair, we next determined the sensitivity of $PARP3^{-/-}$ A549 cells to genotoxins with distinct effects on DNA. $PARP3^{-/-}$ A549 cells were not significantly sensitive to IR (Fig. 4a), as previously shown[25] and were only slightly sensitive to etoposide and bleomycin at higher concentrations (Fig. 4b,c), as previously shown[21].

In contrast, $PARP3^{-/-}$ cells were profoundly sensitive to pyridostatin (Fig. 4d) and PhenDC3 (Supplementary Fig. 4h), small molecules that stabilize G quadruplex (G4) DNA[26,27]. G4 DNA is a secondary structure that can form in single-stranded guanine-rich DNA sequences. G4 DNA structures arise when four guanine bases interact through non-Watson–Crick base pairing to form planar G-tetrads that can stack into thermodynamically stable structures[28]. Estimates of the occurrence of G4 DNA in the human genome range from ~375,000 from computational predictions[29–31] to >700,000 experimentally observed G4 DNA structures[32]. G4 DNA can obstruct replication and repair[33–36]. In fact, pyridostatin leads to cell cycle arrest by inducing replication- and transcription-dependent DNA damage that preferentially, but not exclusively, occurs within gene bodies enriched for G-quadruplex forming sequences[37]. Although pyridostatin is an imperfect tool for studying G4 DNA, two recent studies have reported clear enrichments for G4 motifs at sites of pyridostatin-induced damage[37] and pyridostatin binding[32]. Therefore, we used pyridostatin as a tool to preferentially target regions that are enriched for G4 DNA. Furthermore, the observation that $PARP3^{-/-}$ cells are also sensitive to PhenDC3, a structurally distinct G4 DNA ligand, lends support to the hypothesis that

$PARP3^{-/-}$ cells are preferentially susceptible to stabilization of G4 DNA.

At a concentration of 2.5 µM, pyridostatin reduced colony formation of wild-type A549 cells by <1 log$_{10}$ but completely eliminated colony formation of $PARP3^{-/-}$ cells (Fig. 4d). The pyridostatin sensitivity of $PARP3^{-/-}$ cells was complemented by re-expression of PARP3 (Supplementary Fig. 4i,j). Knockdown of PARP3 with short interfering RNA (siRNA) resulted in

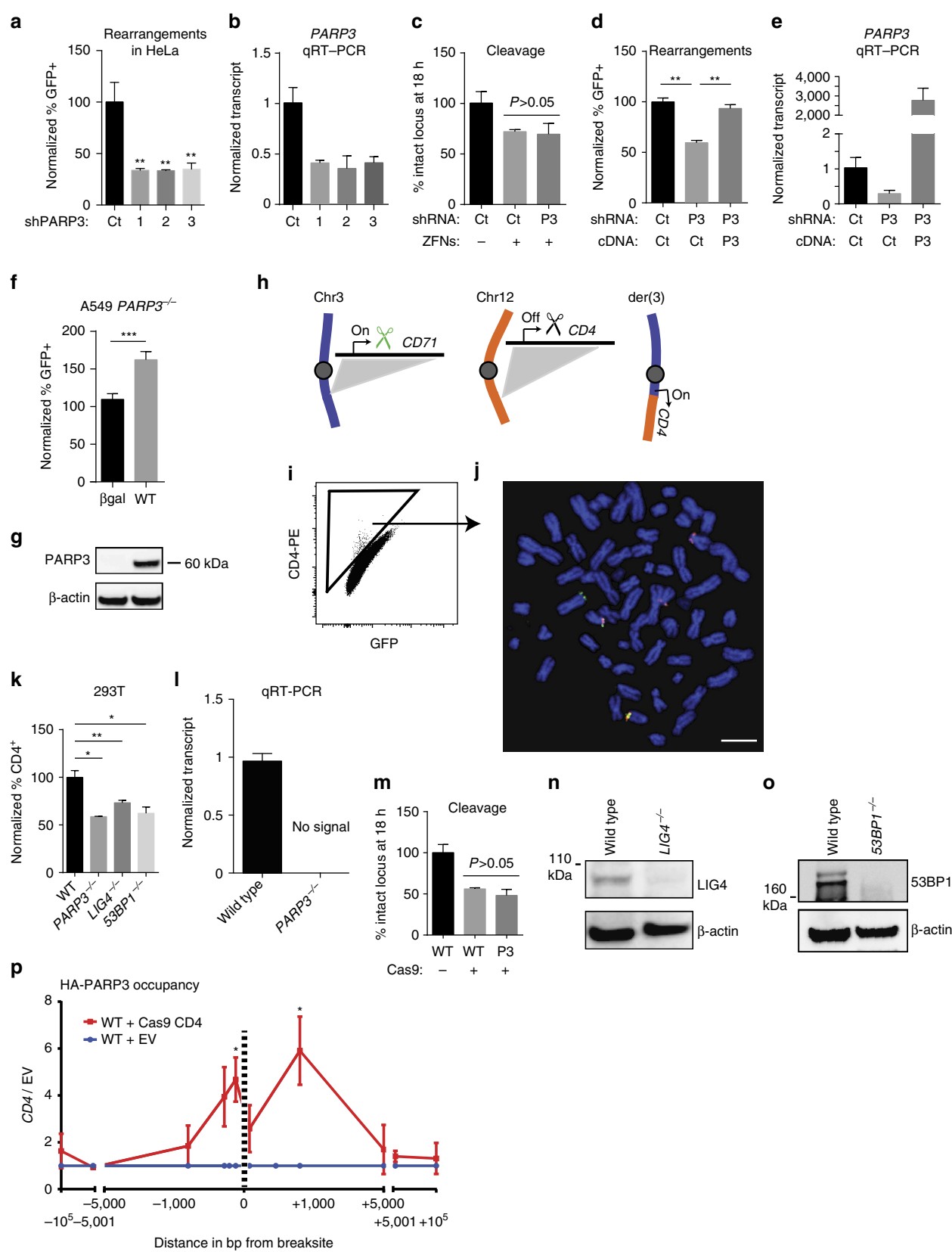

pyridostatin sensitivity that was similar to the $PARP3^{-/-}$ cells (Fig. 4e; Supplementary Fig. 4k). In contrast, siRNA knockdown of either PARP1 or PARP2 did not increase sensitivity to pyridostatin (Fig. 4e).

A recent study demonstrated that PARP1 can influence transcription[24], raising the possibility that the DNA repair abnormalities we observed are related to alterations in transcription from loss of PARP3. To agnostically assess transcriptional effects, we performed RNA-sequencing (RNA-seq) of wild-type and $PARP3^{-/-}$ cells in the presence or absence of pyridostatin. We found that only 61 genes were significantly differentially expressed between wild-type and $PARP3^{-/-}$ A549 cells in the absence of pyridostatin (Supplementary Table 1; Supplementary Fig. 5). We performed gene ontology enrichment analysis[38,39] and KEGG pathway analysis[40,41] using this 61 gene signature, which did not identify any pathways that would explain the observed sensitivity (Supplementary Fig. 6a). Next, we compared the differentially expressed genes in the pyridostatin-treated wild-type and $PARP3^{-/-}$ cells, which identified only 64 differentially expressed genes (Supplementary Table 4; Supplementary Fig. 7). These genes were highly overlapping with those identified in the absence of pyridostatin, suggesting relatively minor differences in the transcriptional response to pyridostatin across genotypes. Again, neither gene ontology enrichment analysis nor KEGG pathway analysis of differentially expressed genes following pyridostatin treatment revealed any pathways that might explain the observed phenotype (Supplementary Figs 6b and 8). Thus, and in contrast with PARP1, PARP3 does not appear to serve a major regulatory role in transcription, at least within our experimental system.

To test whether the effect on pyridostatin resistance depends on PARP3 catalytic activity, we treated wild-type cells with ME0328, a PARP3-specific inhibitor that inhibits PARP3 with sevenfold greater potency compared with PARP1 (ref. 19). ME0328 (3 µM) did not elicit any cytotoxicity (Supplementary Fig. 9a), but the same concentration sensitized cells to pyridostatin (Fig. 4f). Treatment with the PARP inhibitor KU0058948 at 500 nM concentration (half-maximal inhibitory concentration for PARP1 3.4 nM; PARP2 ~6 nM, PARP3 85 nM)[42,43] conferred the same sensitization to pyridostatin as ME0328 (Fig. 4f). Taken together, these results suggest a unique function for PARP3 in the regulation of G4 DNA.

The BLM helicase is known to bind and unwind G4 DNA *in vitro*[44–47]. In addition, BLM facilitates replication through G4 DNA in telomeric sequences *in vivo*[48]. However, the role of BLM in promoting DSB repair specifically within regions that harbour abundant G4 DNA remains unclear. On the basis of the *in vitro* data, one possibility is that BLM facilitates the repair of DSBs within G4-rich regions by unwinding secondary DNA structures. We examined the recruitment of BLM to a CRISPR-CAS9-targeted DSB in the *CD4* locus by ChIP. We selected the *CD4* locus because two different *in silico* algorithms (QGRS and

Quadbase2) predicted that the *CD4* locus contains abundant guanine-rich repeats predicted to form G4 DNA (Fig. 4g)[49–51]. Furthermore, a genome-wide sequencing-based study reported a high concentration of G4 DNA at this locus (Fig. 4g)[32].

After induction of a DSB at the *CD4* locus, BLM recruitment was significantly increased in $PARP3^{-/-}$ cells by ChIP relative to wild-type cells (Fig. 4h). One possible explanation for this finding is that BLM is recruited to unwind G4 DNA that is more abundant in $PARP3^{-/-}$ cells. We validated the ChIP signal given by the BLM antibody by immunoprecipitation (IP) followed by western blot (Supplementary Fig. 9b) and by using an siRNA specificity control (Supplementary Fig. 9c). Consistent with the finding at a single DSB, overall chromatin-bound BLM was increased in $PARP3^{-/-}$ cells in the absence of exogenous damage (Supplementary Fig. 9f,g). If recruitment of BLM in $PARP3^{-/-}$ cells is dependent on the presence of G4 DNA, it would follow that PARP3 status would not influence the recruitment of BLM to a CRISPR-CAS9-targeted DSB at a locus with minimal G4 DNA. In contrast to the *CD4* locus, the oestrogen receptor 1 (*ESR1*) locus is predicted and observed to contain relatively little G4 DNA (Fig. 4i). As predicted, we found that there was no significant difference between the recruitment of BLM to a DSB in *ESR1* between wild-type and $PARP3^{-/-}$ cells (Fig. 4j).

A recent study described hf2, an engineered single-chain antibody specific for G4 DNA[52]. To validate hf2 at sites of enriched G4 DNA in our experimental system, we tested loci previously characterized for G4 DNA content: (1) telomeric DNA features the most abundant G4 DNA in the cell, (2) the *MYC* promoter is known to contain abundant G4 DNA[52,53] (Supplementary Fig. 9h), as mentioned above, (3) the *ESR1* enhancer locus[37] (Fig. 4i) and two loci on chromosomes 8 and 22 that are predicted and observed to contain very little G4 DNA (Supplementary Fig. 9i,j). Following IP with hf2 using pyridostatin-treated 293T cells, we observed no enrichment at the *ESR1*, chr8 and chr22 loci, but significant enrichments at the *MYC* promoter and at telomeres in the absence of DNA damage (Fig. 4k). We therefore measured the G4 DNA content at the *CD4* locus following induction of a targeted DSB following treatment with pyridostatin. G4 DNA was enriched to a greater extent in $PARP3^{-/-}$ cells compared to wild-type cells at sites flanking the DSB (Fig. 4l). In contrast, at the *ESR1* locus, we did not observe any enrichment of G4 DNA in $PARP3^{-/-}$ cells following a targeted DSB (Supplementary Fig. 10a). These data are consistent with a role for PARP3 in negatively regulating G4 DNA induced by a DSB.

**$PARP3^{-/-}$ cells are susceptible to pyridostatin-induced DSBs.** On the basis of the increased sensitivity of $PARP3^{-/-}$ cells to pyridostatin, we hypothesized that ligand-stabilized G4 structures would result in more extensive DSBs. To test this, we performed immunofluorescence (IF) to quantify γH2AX and 53BP1 foci following pyridostatin treatment. $PARP3^{-/-}$ cells exhibited significantly more γH2AX (Fig. 5a) and 53BP1 foci (Suppleme-

**Figure 3 | PARP3 promotes chromosomal rearrangements in several human cell types.** (**a**–**c**) Normalized frequency of rearrangements (**a**) compared to control (Ct), *PARP3* transcript levels (**b**) and cleavage by the *AAVS1* ZFNs measured by quantitative PCR across the targeted site (**c**) in HeLa cells transduced with shRNA targeting *PARP3*. P3, PARP3. (**d**,**e**) Normalized frequency of rearrangements (**d**) and qRT–PCR (**e**) in HeLa ZITR cells transduced with lentivirus expressing control (Ct) shRNA or shRNA targeting PARP3 3′-untranslated region (P3) and expressing control (Ct) cDNA or PARP3 cDNA. (**f**,**g**) Normalized frequency of rearrangements (**f**) and immunoblots (**g**) in $PARP3^{-/-}$ A549 cells with adenovirus-mediated re-expression of wild-type (WT) PARP3. (**h**–**j**) Schematic of CRITR assay for translocations between *CD71* and *CD4* loci (**h**), flow cytometry (**i**) and metaphase fluorescence *in situ* hybridization (**j**) of CD4$^+$ 293T cells 48 h after transfection of CAS9 and gRNAs. Green probe, RP11-436M6, *CD71*. Red probe, RP11-277E18, *CD4*. Scale bar, 10 µm. (**k**–**o**) Normalized frequency of rearrangements using CRITR assay in wild-type (WT), $53BP1^{-/-}$, $PARP3^{-/-}$ and $LIG4^{-/-}$ 293T cells (**k**) with qRT–PCR of *PARP3* (**l**), immunoblots of indicated proteins (**m**,**n**) and intact locus after expression of CAS9 and gRNA measured by quantitative PCR across the targeted site (**o**). HA-PARP3 occupancy at *CD4* 18 h after transient expression of CAS9 and gRNA by ChIP, represented as ratio of CAS9 with *CD4*-directed gRNA to CAS9 with empty vector (EV) (**p**). bp, base pairs. Data are presented as mean ± s.e. of $n = 3$. P values were calculated using unpaired Student's t-test. *$P < 0.05$, **$P < 0.01$, ***$P < 0.001$.

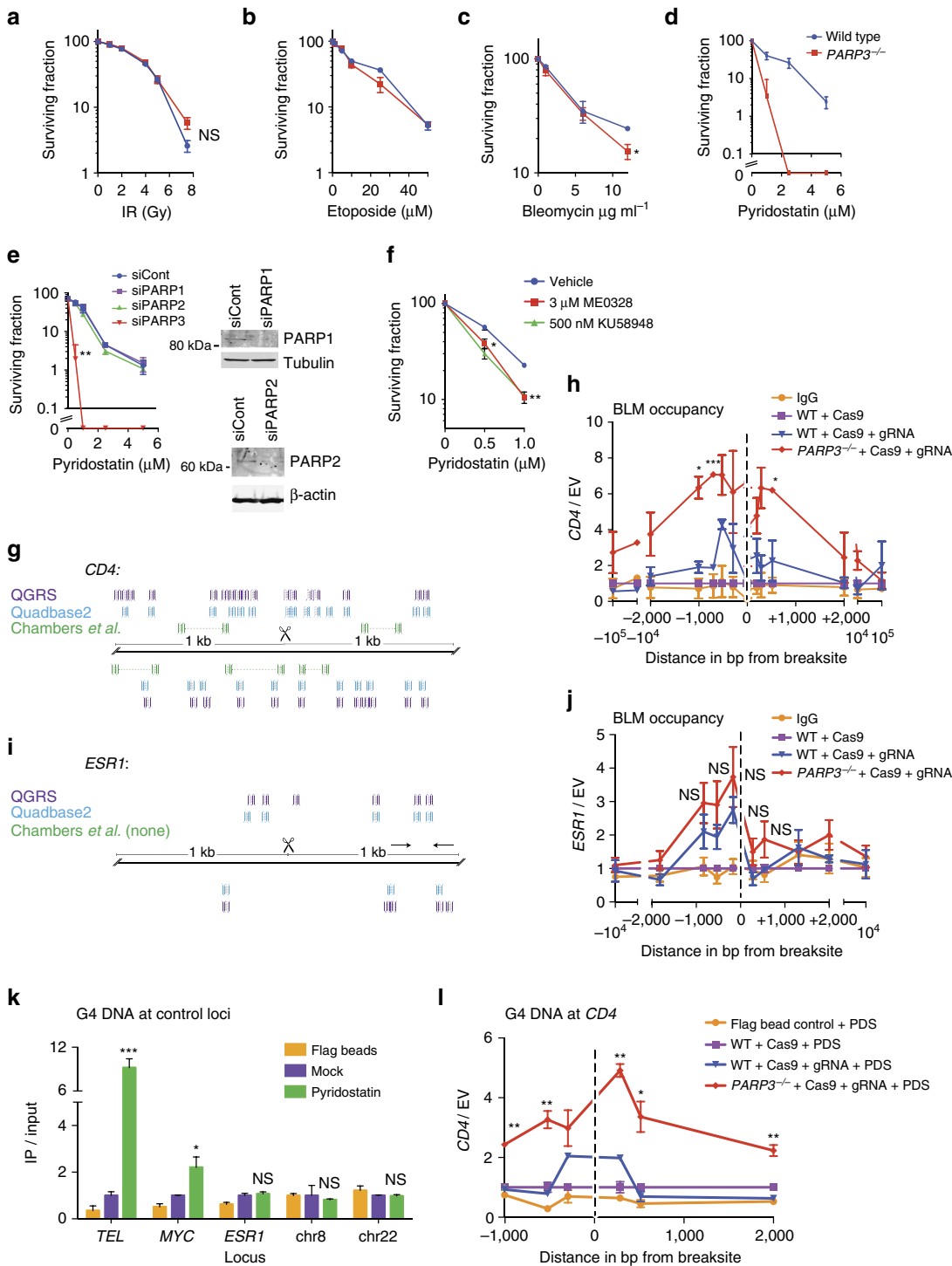

**Figure 4 | PARP3$^{-/-}$ cells are sensitive to the stabilization of G4 DNA** (**a**–**d**) Colony survival assays using wild-type and PARP3$^{-/-}$ A549 cells with the indicated doses of ionizing radiation (**a**), etoposide (**b**), bleomycin (**c**) and pyridostatin (**d**). (**e**) Colony survival assay in wild-type A549 cells with siRNA for control (siCont), PARP1, PARP2 or PARP3 with corresponding immunoblots. (**f**) Colony survival assay in wild-type A549 cells treated with vehicle, 3.0 μM ME0328 or 500 nM KU58948 and pyridostatin. (**g**) Schematic of sequences predicted by QGRS (purple) and Quadbase2 (blue) and experimentally observed to form G4 DNA by Chambers et al.[32] (green) in a 2,000 bp window surrounding the CD4 CRISPR-CAS9 gRNA. Scissors, CRISPR-CAS9 cut site. (**h**) BLM occupancy at the CD4 locus measured by ChIP 18 h after transient expression of CAS9 alone or with gRNA targeting CD4. (**i**) Schematic of sequences G4 DNA sequences depicted as in **g** in a 2,000 bp window surrounding ESR1 CRISPR-CAS9 gRNA. Scissors, CRISPR-CAS9 cut site. (**j**) BLM occupancy at the ESR1 locus measured by ChIP 18 h after transient expression of CAS9 alone or with gRNA targeting CD4. (**k**) G4 DNA content measured by immunoprecipitation (IP) with hf2 antibody following 24 h treatment with vehicle or pyridostatin (PDS) at the telomere (TEL), MYC promoter (MYC), ESR1 enhancer, chr8 or chr22 loci. (**l**) Quantification of G4 DNA content in the indicated genotypes at the CD4 locus measured by IP with hf2 antibody 24 h after pyridostatin (PDS) treatment and 18 h after transfection with CRISPR-CAS9 with or without gRNA for CD4. bp, base pairs; PDS, pyridostatin. Data are mean ± s.e. of n = 3. P values calculated using unpaired Student's t-test. *P < 0.05, **P < 0.01, ***P < 0.001. NS, not significant.

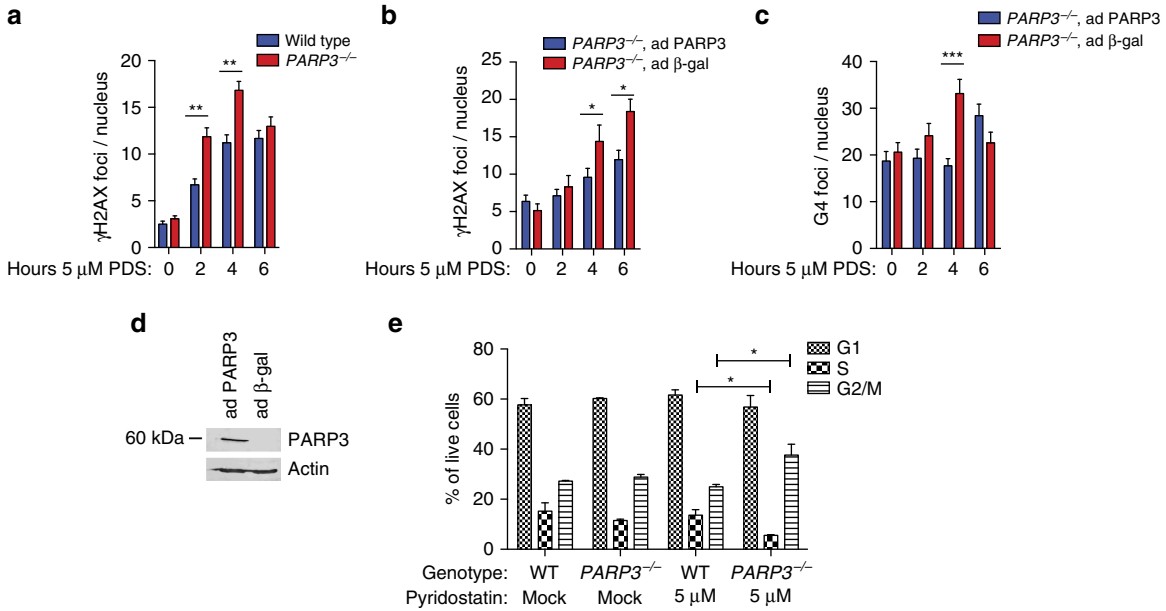

**Figure 5 | *PARP3*$^{-/-}$ cells are more susceptible to pyridostatin-induced DSBs.** (**a**) Quantification of immunofluorescence (IF) for γH2AX at the indicated time points following treatment with 5 μM pyridostatin (PDS) in wild-type and *PARP3*$^{-/-}$ A549 cells. (**b,c**) Quantification of IF for γH2AX (**b**) and G4 DNA using the 1H6 antibody (**c**) at the indicated time points, and immunoblots (**d**) in *PARP3*$^{-/-}$ A549 cells infected with adenovirus-expressing PARP3 (ad PARP3) or β-galactosidase (ad β-gal). (**e**) Cell cycle dynamics for wild-type (WT) or *PARP3*$^{-/-}$ A549 cells mock treated or treated with 5 μM pyridostatin. Data are presented as mean ± s.e. of *n* = 3. *P* values calculated using unpaired Student's *t*-test. A two-way analysis of variance (alpha set at 0.05) gave the following *P* values for cell cycle phases: G1, *P* = 0.5955, S, *P* = 0.0152, and G2, *P* = 0.0111. *$P < 0.05$, **$P < 0.01$, ***$P < 0.001$.

ntary Fig. 10b,c) at earlier time points with a high degree of co-localization between damage markers (Supplementary Fig. 11). This effect was reversed by ectopic expression of PARP3 (Fig. 5b,d; Supplementary Figs 10d,e and 12). To directly measure the impact of PARP3 on the abundance of G4 DNA, we made use of the 1H6 antibody, a reagent recently shown to visualize G4 DNA by IF[54]. Expression of PARP3 reduced G4 foci in *PARP3*$^{-/-}$ cells treated with pyridostatin based on foci stained with 1H6 (Fig. 5c,d; Supplementary Fig. 13).

After extended incubations, pyridostatin-mediated DNA DSBs are known to trigger the G2/M checkpoint[37]. Incubation of wild-type cells for 24 h with 5 μM of pyridostatin did not significantly alter cell cycle dynamics but significantly reduced the S-phase fraction in *PARP3*$^{-/-}$ cells with a concomitant increase in the G2/M fraction (Fig. 5e), consistent with enhanced induction of the checkpoint.

**PARP3 promotes CtIP and RPA deposition at DNA DSBs.** We hypothesized that the increased prevalence of G4 structures in PARP3-deficient cells would suppress binding by the endonuclease CtIP, which converts DSB ends into single-stranded DNA tails that are intermediates for HR. In fact, the efficiency of HR at a single DSB was reduced in PARP3-depleted cells containing a HR reporter (Supplementary Fig. 10f–h), consistent with a previous report[21]. Cutting by CAS9 at the *CD4* locus was similar in wild-type and *PARP3*$^{-/-}$ cells (Fig. 6a), but *PARP3*$^{-/-}$ cells had less CtIP accumulation flanking the DSB (Fig. 6b). The ChIP signal given by the CtIP antibody was validated by IP (Supplementary Fig. 9d) and siRNA specificity control (Supplementary Fig. 9e). Processing by CtIP facilitates deposition of the single-stranded DNA-binding protein RPA. *PARP3*$^{-/-}$ cells also had reduced RPA deposition around the targeted DSB (Fig. 6c) but significantly increased γH2AX signal (Fig. 6d). The latter finding is consistent with a report that

depletion of PARP3 leads to the persistence of γH2AX foci[23]. In contrast, we did not observe any significant PARP3-dependent differences in CtIP or RPA deposition or γH2AX signal at a targeted DSB in *ESR1* (Supplementary Fig. 14a–c), where little or no G4 DNA is expected to form.

To further support these findings, we examined RPA occupancy following targeted DSBs at six additional loci: three in G4-rich regions and three in G4-poor regions (Supplementary Fig. 15). We observed that RPA deposition was significantly reduced in *PARP3*$^{-/-}$ cells near DSBs in G4-rich regions (Supplementary Fig. 15a–c,g–i) but not affected by PARP3 deficiency in G4-poor regions (Supplementary Fig. 15d–e,j–l).

To confirm these findings using an orthogonal method, we performed IF microscopy for RPA at 2 h after 10 Gy IR. Compared to wild-type cells, fewer *PARP3*$^{-/-}$ cells had both γH2AX and RPA foci (Fig. 6e,f). We observed the same reduction in *PARP3*$^{-/-}$ cells with double-positive foci after synchronizing the cells in S-phase by serum starvation (Fig. 6g; Supplementary Fig. 16a). Thus, the reduction in RPA foci was not attributable to alterations in cell cycle. Together, these data suggest that PARP3 deficiency inhibits CtIP and RPA deposition at DSBs in G4-rich regions, perhaps leading to a delay in repair as evidenced by persistent γH2AX signal.

**PARP3 and BLM cooperate to promote NHEJ.** Previous studies have demonstrated that PARP3 loss reduces the efficiency of NHEJ[21], which delays the repair of IR-induced DNA damage[23]. We confirmed the effect of PARP3 depletion on NHEJ between targeted DSBs (Supplementary Fig. 16b–d)[55]. Because G4 DNA can be a substrate for BLM *in vitro* and at telomeres[46,48], we reasoned that loss of BLM in PARP3-deficient cells might further suppress the resolution of G4 DNA structures, which could reduce both HR and NHEJ, leading to the persistence of DSBs.

We used siRNA to deplete BLM in wild-type and *PARP3*$^{-/-}$ cells, and measured the kinetics of IR-induced 53BP1 and

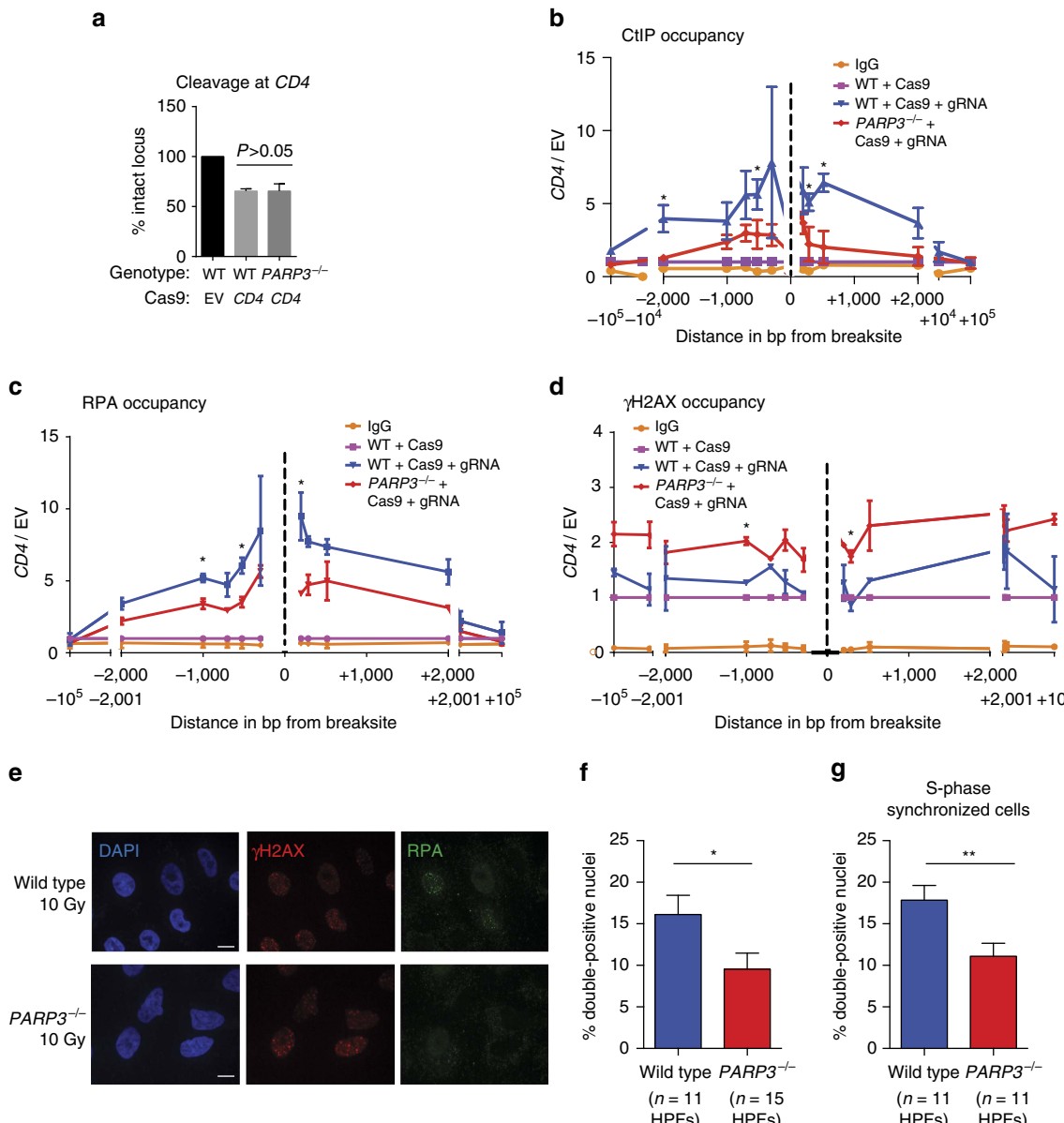

**Figure 6 | PARP3 promotes deposition of CtIP and RPA at DNA DSBs.** (**a**) CAS9-mediated cleavage at *CD4* measured by quantitative PCR across the targeted site 18 h after transient expression of CAS9 and gRNA targeting *CD4* (CD4) or CAS9 with empty vector (EV). (**b–d**) CtIP (**b**), RPA (**c**) and γH2AX (**d**) occupancy 18 h after transient expression of CAS9 alone or with gRNA targeting *CD4*. bp, base pairs. Data are represented as mean ± s.e. of $n \geq 2$ biological replicates. *P* values calculated using unpaired Student's *t*-test. *$P < 0.05$, **$P < 0.01$, ***$P < 0.001$. (**e,f**) Immunofluorescence for γH2AX and RPA (**e**), and quantification of γH2AX and RPA double-positive cells (**f**) at 2 h following 10 Gy IR. Scale bars, 10 μm. (**g**) Quantification of γH2AX and RPA double-positive cells at 24 h after release from serum starvation and 2 h after 10 Gy IR. Data are represented as per cent nuclei per high-power field (HPF). Two-sided *t*-test was calculated on three biological replicates with a representative replicate shown.

γH2AX foci. Deficiency of either PARP3 or BLM decreased the formation of 53BP1 foci between 1 and 6 h after 10 Gy IR (Fig. 7a,c). Knockdown of BLM in *PARP3*$^{-/-}$ cells further suppressed 53BP1 focus formation after IR (Fig. 7a,c). Knockdown of BLM in *PARP3*$^{-/-}$ cells also resulted in significantly more γH2AX foci as early as 1 h following 10 Gy compared to cells lacking BLM or PARP3 alone (Fig. 7b,c). In wild-type cells and in cells lacking either BLM or PARP3, 40% or more of γH2AX and 53BP1 were co-localized, compared to <10% of foci in *PARP3*$^{-/-}$ cells with BLM knockdown (Supplementary Fig. 16e). Together, these data support a model (Fig. 8), in which BLM and PARP3 cooperate to regulate G4 DNA structures and promote DSB repair.

## Discussion

We performed a medium-throughput screen for genetic factors that modulate the formation of chromosomal rearrangements. We identified and validated two genes that suppress (*UBC9* and *RAD50*) and three genes that promote (*53BP1*, *DDB1* and *PARP3*) chromosomal rearrangements. The results of our screen are useful both for significantly expanding the catalogue of factors involved in rearrangements in human cells and for demonstrating the feasibility of genetic screening for factors involved in rare chromosomal events. On the basis of our stringent definition of 'hits', all five factors validated in an alternate cell lineage, but the stringent definition likely resulted in multiple false-negatives. Importantly, we intentionally utilized shRNA-based knockdown

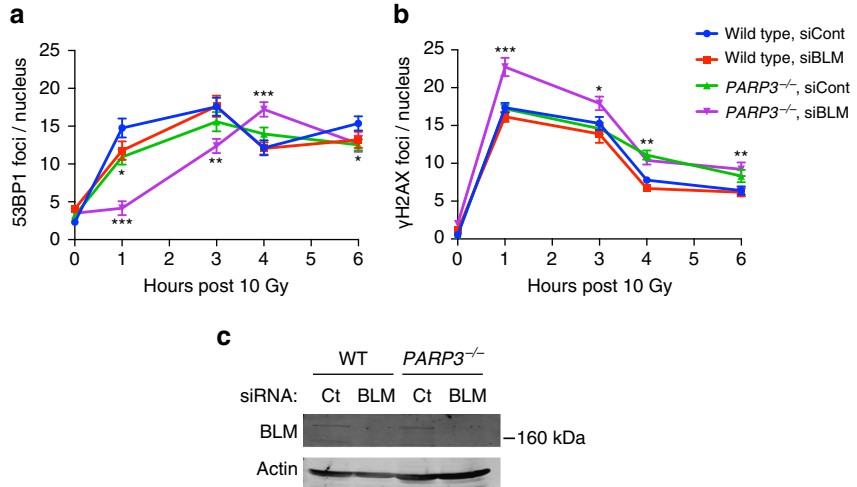

**Figure 7 | PARP3 and BLM cooperate to repair DNA DSBs.** (**a–c**) Quantification of IF for 53BP1 (**a**) and γH2AX (**b**) at the indicated time points in wild-type (WT) or *PARP3*$^{-/-}$ A549 cells transfected with siRNA targeting control (siCont) or BLM (siBLM). Data are represented as mean ± s.e. of three biological replicates. *P* values calculated using unpaired Student's *t*-test. *$P < 0.05$, **$P < 0.01$, ***$P < 0.001$. (**c**) immunoblot after transfection with siRNA targeting control (Ct) or BLM.

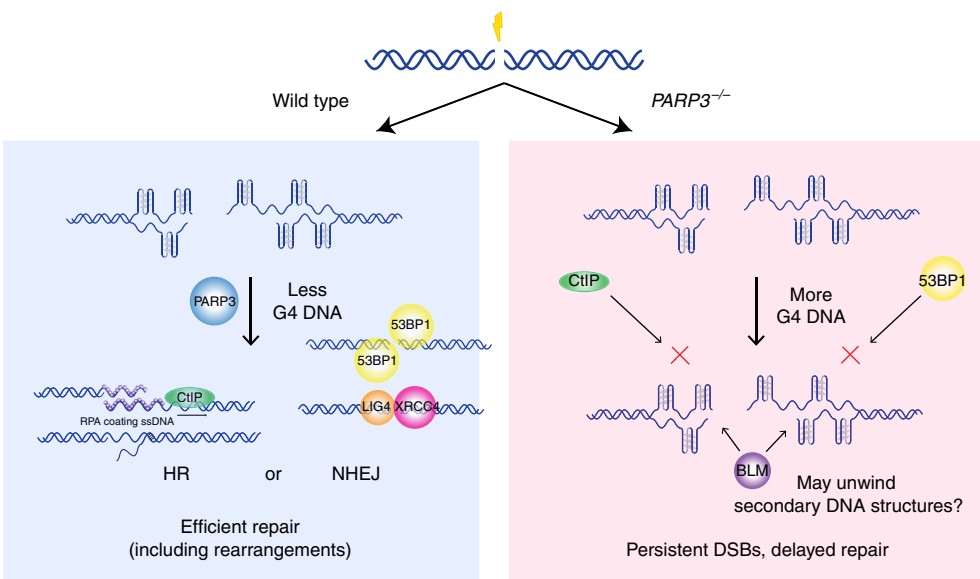

**Figure 8 | Model of PARP3 regulation of G4 DNA at DNA DSBs.** PARP3 negatively regulates G4 DNA after DNA DSBs to facilitate repair of the damage (left). In the absence of PARP3, G4 DNA accumulates at sites of damage and delays repair of DSBs by preventing deposition of repair factors.

rather than CRISPR-based knockout for the screen, as this allowed for the identification of factors required for viability (for example, DDB1) and the assessment of dosage effects.

We examined PARP3 in detail and observed that, as previously reported, PARP3-deficient cells had reduced efficiencies of both HR and NHEJ but relatively little or no sensitivity to traditional DSB-inducing agents[21,23] (Fig. 4a–c; Supplementary Figs 10g,h and 16c,d). This creates a conundrum, as cells must presumably either repair DSBs, arrest or undergo apoptosis. A key insight came from the finding that PARP3-deficient cells are highly sensitive to pyridostatin and PhenDC3, small molecules that preferentially but not exclusively stabilize and damage regions of G4 DNA. This raised the possibility that PARP3-deficient cells are susceptible to G4 DNA-mediated damage. In fact, *PARP3*$^{-/-}$ cells had increased G4 DNA compared to wild-type cells at sites flanking DSBs within G4-rich regions. Taken together, these findings suggest that *PARP3*$^{-/-}$ cells

accumulate more G4 DNA than wild-type cells. Of note, the EJ5 (ref. 55) and DR-GFP[56] reporters used to measure NHEJ and HDR, respectively, contain high concentrations of sequence predicted to form G4 DNA (Supplementary Fig. 16f,g).

We propose a model in which the existence of G4 DNA delays but does not prevent the repair of DSBs in the absence of PARP3 (Fig. 8). There are several aspects of the model that require experimental clarification: (1) As discussed above, the specificity of G4 DNA ligands is imperfect[37]. Therefore, it will be important to conclusively define the mechanism of sensitization to pyridostatin and PhenDC3 in the absence of PARP3. The RNA-seq experiments described in Supplementary Figs 5–8 represent an important first step but further experiments are needed. (2) The role of BLM helicase at DNA DSBs within G4-rich regions remains unclear. We have shown that BLM accumulates to a greater extent in the absence of PARP3 at a DNA DSB targeted to a region of abundant G4 DNA

potential. Given that BLM has been shown to unwind G4 DNA *in vitro*[46], one possibility is that BLM unwinds G4 DNA in the context of DSBs *in vivo* as well. However, several other functions for BLM have been reported during repair of DSBs including unwinding Holliday junctions[57], promoting DNA DSB resection by recruiting EXO1 and DNA2 (ref. 58), and blocking recombination by disrupting RAD51 nucleoprotein filaments[59,60]. Further experiments are required to distinguish between these mechanistic possibilities. (3) Our data showing PARP3-dependent differences in accumulation of G4 DNA and BLM helicase at DNA DSBs is based on profiling of two targeted loci: one with high G4 DNA potential (*CD4*) and one lacking the potential to form G4 DNA structures (*ESR1*). Altogether, our data showing PARP3-dependent differences in RPA accumulation assessed eight loci with varying G4 DNA potential; however, this remains a relatively small set of loci. Therefore, to strengthen our model and overcome potential artefacts resulting from comparison of a small number of loci, data from a larger set of loci with varying G4 DNA content is needed. Techniques that will allow profiling of the PARP3 dependency of G4 DNA in a genome-wide manner were recently reported[61]. The combination of these techniques with a larger set of targeted DNA DSBs will provide important future data to refine or even refute our model. (4) Further experiments are needed to determine the extent to which PARP3 plays a role in G4 DNA resolution in the absence of damage, at sites of damage other than DSBs (for example, interstrand crosslinks), and in species other than humans.

Recent studies have begun to shed light on the role of chromatin structure in G4 DNA formation. For example, ATRX, a member of the SWI–SNF nucleosome remodelling complex, binds specifically to G4 DNA[62]. A recent study identified PARP3-specific targets of mono(ADP)ribosylation, including several histones and chromatin-bound factors; among the latter was ATRX[24]. One possible hypothesis generated by our data is that PARP3 modifies chromatin structure to establish an environment that limits G4 DNA.

As previously reported, PARP3-deficient cells were not sensitized to IR but were partially sensitized to bleomycin[20,21]. Bleomycin has been shown to preferentially induce DSBs near DNA secondary structures[63,64] and in particular in telomeric regions[65]. Therefore, bleomycin could be selectively targeting DNA DSBs to regions with abundant G4 DNA, and thereby confer a slight sensitivity in PARP3-deficient cells.

Several additional questions remain unanswered by our findings. First, are the suppression of NHEJ and binding of CtIP and RPA by PARP3 loss mechanistically linked to the lower frequency of chromosomal rearrangements in cells lacking PARP3? Although significant, the reduced binding of RPA at the targeted break in the *CD4* locus in *PARP3*$^{-/-}$ cells was modest and thus bears further exploration in other systems. How does PARP3 catalytic and/or scaffold function mediate the suppression of G4 structures on chromatin in the presence of DNA damage? Finally, does PARP3 inhibition have a therapeutic role, either alone or in combination with G4-stabilizing therapeutics? In fact, G4 DNA structures are enriched in proto-oncogenes such as *MYC*[53,66], suggesting the potential for simultaneously targeting of DNA damage and modulating oncogene expression. Small molecules that bind G4 structures have shown promise in early studies[34,67,68]. Certain malignancies, including those of the stomach and liver, have abundant G4 DNA relative to the surrounding stromal tissue[69] and may therefore be ideal targets for G4 DNA ligands. Selective PARP3 inhibitors could represent a tractable strategy to combine with G4 DNA ligands to exacerbate G4 DNA formation, DSB induction and tumour cell death.

## Methods

**Cell culture and reagents.** A549, HeLa, 293T and U2OS cells were cultured in DMEM (Gibco) supplemented with 10% fetal bovine serum (FBS), and 50 U ml$^{-1}$ penicillin and 50 ng ml$^{-1}$ streptomycin (Gibco). None of the cell lines used in this study are listed in the database of commonly misidentified cell lines maintained by the International Cell Line Authentication Committee[70]. The A549 cell line was a gift from the Bradner Lab at Dana-Farber Cancer Institute and was verified by short tandem repeat (STR) profiling. The HeLa and U2OS cells originated from the Jasin Lab at Memorial Sloan Kettering Cancer Center. The 293T cells were gifts from the Alt Lab at Boston Children's Hospital. Cells were tested for mycoplasma contamination by the IDEXX Laboratories or using the Lonza Mycoplasma kit. Puromycin (Gibco) selection in A549 and 293T cells was at 1.0 µg ml$^{-1}$. Hygromycin B (Invitrogen) selection in A549 cells was at concentration (200 µg ml$^{-1}$). Cell synchronization of A549 cells was achieved by growing cells to confluence, reducing FBS to 0.1% for 72 h and releasing cells into DMEM with 20% FBS. A549 cells were treated at the indicated concentrations of pyridostatin (Sigma Aldrich), PhenDC3 (Polysciences), ME0328 (Selleck), KU0058948 (Axon MedChem) or Bleomycin (Santa Cruz Biotechnology) when indicated. All cells were cultured under normal oxygen conditions (21% O$_2$, 5% CO$_2$, 37 °C).

**Preparation of adenovirus.** Complementary DNAs (cDNAs) encoding AAVS1 ZFNs 1 and 2 separated by a ribosomal insertion site, AAVS1 ZFN 1 alone, or PARP3 were subcloned into pAd/CMV (Invitrogen) using Gateway cloning (Invitrogen). The resulting shuttle vectors were transfected into 293A cells (Invitrogen) to generate recombinant adenovirus. Adenovirus was concentrated by centrifugation in a caesium chloride gradient. A549 and HeLa cells were routinely infected with 2 × 10$^9$ plaque-forming unit (p.f.u.) ml$^{-1}$ adenovirus. The I-Sce*I* adenovirus was a gift from R. Hromas (University of Florida). As a control for adenoviral infection, cells were infected with Ad-β-galactosidase.

**Constructs.** The AAVS1 ZFNs were a kind gift of Fyodor Urnov. The PARP3 WT was previously described[23]. The PARP3 WT cDNA was subcloned in pCAGGS using XbaI and XhoI restriction sites. PARP3 WT was cloned into pAd Easy using Gateway Cloning (Invitrogen).

**Construction of rearrangement reporter.** Enhanced-GFP open reading frame or human CD4 with a truncated cytoplasmic domain (pMACS 4.1, Miltenyi Biotec) were cloned downstream of the AAVS1 zinc-finger recognition site, a 300 bp mouse intronic spacer sequence and the 2A ribosomal stutter site. A hygromycin resistance gene driven by a pgk promoter was cloned downstream of the transgene open reading frame (Supplementary Fig. 1a). A549 or HeLa cells were transiently transfected with the reporter cassettes and selected in hygromycin (200 µg ml$^{-1}$) for 14 days to allow clones to grow out. Individual clones were amplified and tested to ensure transgene negativity. Finally, clones were evaluated for induction of transgene expression upon delivery of the AAVS1 ZFNs.

**Rearrangement factor screen.** shRNAs targeting DNA repair factors and controls were obtained from The RNAi Consortium (http://www.broadinstitute.org/rnai/trc) as pLKO lentiviruses (*n* = 966, see Supplementary Table 1 for clone ID# and target sequences). A549 cells containing either a randomly integrated *GFP* or *CD4* rearrangement reporter were plated at 2,500 cells per well of a 96-well plate in 100 µl of DMEM with 10% FBS with 10 µg ml$^{-1}$ polybrene. A measure of 2.0 µl lentiviral supernatant was added and the plate was centrifuged at 300*g* for 30 min, and then placed in a 37 °C incubator for 24 h. Cells were trypsinized and pooled, 10$^6$ cells were saved for input shRNA analysis, and the remaining cells were amplified for 7 days in DMEM with 10% FBS containing 1.0 µg ml$^{-1}$ puromycin. On day 7, cells were infected with adenovirus-expressing AAVS1 ZFNs at 3 × 10$^9$ p.f.u. ml$^{-1}$ for 48 h at which point cells were sorted into transgene-positive and transgene-negative populations using a FACS Aria II (BD Bioscience). A total of 200,000 transgene-positive cells (and an equivalent number of transgene-negative cells) were collected for each replicate to ensure that each shRNA in the library was represented by at least 200 transgene-positive cells. Genomic DNA was collected from the two populations (Qiagen QIAmp kit). The screen was repeated in *n* = 3 (GFP) and *n* = 4 (CD4) independent biological replicates.

The shRNA encoded in the genomic DNA was amplified using two rounds of PCR. Primary PCR reactions were performed using up to 10 µg of genomic DNA in 100 µl reactions consisting of 10 µl buffer, 8 µl dNTPs (2.5 mM each), 10 µl of 5 µM primary PCR primer mix (see below) and 1.5 µl Takara exTaq. For the secondary PCR amplification, the reaction was performed using modified forward primers, which incorporated Illumina adaptors and six-nucleotide barcodes. Secondary PCR reactions were pooled and run on a 2% agarose gel. The bands were normalized and pooled based on relative intensity. Equal amount of sample was run on a 2% agarose gel and gel purified. Samples were sequenced using a custom sequencing primer on an Illumina Hi-Seq and intensity normalized.

**Lentiviral transductions.** Cells were plated at 60,000 cells per well of 12-well tissue culture plates, and 10 µl of lentivirus and polybrene to a final concentration of 10 µg ml$^{-1}$ were added. Cells were spun at 1,178*g* for 15 min and returned to

incubator overnight. Twenty-four hours after transduction, cells were cultured in 1.0 µg ml$^{-1}$ puromycin for 48 h.

**Flow cytometry.** To avoid cleavage of the *CD4* transgene in the A549 rearrangement reporter cells, Cell Stripper (MediaTech Inc.) was used to dissociate the cells. For FACS analysis of the A549 rearrangement CD4 reporter cell line, CD4-PE (VIT4, Milteyni Biotec) was used at a dilution of 1:500 to stain for 20 min at room temperature.

**Cytogenetics.** Interphase and metaphase fluorescent *in situ* hybridization (FISH) was performed with either a break-apart probe at the *AAVS1* locus (RP11-155P5 and RP11-46I13) for interphase FISH or probes for *CD4* (RP11-277E18) and *CD71* (RP11-436M6) for metaphase FISH. All cytogenetics for this study were performed at the CytoGenomics Core at Brigham and Women's Hospital.

**CRITR assay.** Guide RNAs targeting CD4 (5′-CAGAGGTGTCTTACCCTAG-3′) and CD71 (5′ CTAGCATTGTGATCGATTC-3′) were cloned into pX330 (Addgene 42230), a plasmid that expresses Cas9 and the chimeric guide RNA. Using Lipofectamine 2000, $1 \times 10^6$ 293T cells were transfected with either 2.5 µg of each pX330 plasmid or only one pX330 as a control. Cells were cultured for 48 h, dissociated from the plate using Cell Stripper (Mediatech, Inc.), stained with anti-CD4-PE (M-T466, Miltenyi Biotec) at a dilution of 1:100 and analysed by flow cytometry.

**Construction of knockout cells.** To produce homozygous deletions in 293T and A549 cells using transient expression of Cas9/CRISPR, $2 \times 10^6$ cells were nucleofected (Amaxa, Lonza) with 2.0 µg each of pX330 (Addgene plasmid 42230) containing guides targeting a 200–400 bp sequence in the gene and 0.2 µg of a plasmid constitutively expressing puromycin. Cells were grown 24 h without selection, 48 h with puromycin selection, and subsequently cloned by serial dilution (293T and A549 cells) or allowed to proliferate for three days and colonies were isolated and analysed by PCR for the presence of the deletion. RNA guides can be found in Supplementary Table 5.

**siRNA and cDNA transfection or nucleofection.** Transfection of siRNA was performed with Dharmafect (Dharmacon) according to the manufacturer's instructions. Cells were plate in six-well plates and each well was transfected with 5 µl of Dharmafect reagent and contained a final siRNA concentration of 30 nM in a total volume of 2 ml. siRNA used are as follows. ON-TARGETplus Human PARP3 (10039) siRNA (L-009297-00), ON-TARGETplus Human PARP2 (10038) siRNA (L-010127-02), ON-TARGETplus Human PARP1 siRNA (L-006656-03), ON-TARGETplus Human BLM (641), ON-TARGETplus Human RBBP8 (CtIP) (J-011376-05), siRNA (L-007287-00) and ON-TARGETplus Non-targeting Pool (D-001810-10). Nucleofection of A549 and 293T cells was performed according the manufacturer's instructions (Amaxa, Lonza) using $2 \times 10^6$ cells and 2.0 µg or 20 pmol of siRNA.

**Real-time qRT–PCR.** Total RNA was extracted from cells using the RNeasy Mini Kit (Qiagen). Quantitative PCR with reverse transcription (qRT–PCR) was performed on a CFX96 Real Time System (Bio-Rad) machine using iTaq UniverSYBR Green (Bio-Rad). qRT–PCR conditions were 50 °C for 10 min; 95 °C for 10 min; 39 cycles of 95 °C for 10 s and 60 °C for 30 s. qRT–PCR primers can be found in Supplementary Table 6.

**Western blotting.** Cells were washed with ice-cold PBS and lysed on ice for 30 min with RIPA lysis buffer supplemented with protease inhibitors (Complete Mini, Roche). Protein concentration was determined using the Bradford Protein Assay kit (Bio-Rad) and a calibration standard curve created from bovine serum albumin standards. The samples were prepared for loading by adding $4 \times$ Laemmli sample buffer (Life Technologies) and heating the samples at 90 °C for 10 min. Total proteins were separated by SDS–polyacrylamide gel electrophoresis on 4–12% Bis-Tris gradient gels (Life Technologies). Proteins in the gel were electrophoretically transferred to the nitrocellulose membrane (Bio-Rad) and then the membrane was blocked in 5% milk with Tris-Buffered Saline Triton X-100 buffer (100 mM Tris, pH 7.4, 500 mM NaCl, 0.1% Triton X-100; TBST). Membranes were incubated with primary antibodies in 5% milk in TBST overnight at 4 °C. Horseradish peroxidase-conjugated secondary antibody incubation was performed for 2 h at room temperature in 5% milk in TBST and signals were visualized by enhanced chemiluminescence (ECL) (Perkin Elmer). Whole portions of western blots are included in Supplementary Fig. 17.

**Immunofluorescence.** At the indicated times after irradiation, cells were pre-extracted with 25 mM HEPES, pH 7.4, 50 mM NaCl, 1 mM EDTA, 3 mM MgCl$_2$, 300 mM sucrose and 0.5% Triton X-100 for 5 min on ice. Cells were fixed with 4% paraformaldehyde for 12 min at room temperature, washed twice with PBS with 1% FBS for blocking and co-stained for 16 h at 4 °C with γH2AX S139

(20E3, Cell Signaling) and RPA32 (A300-244A, Bethyl Labs) at dilutions of 1:500, mouse anti-DNA G quadruplex (G4), (1H6, EMD Millipore; dilution of 1:500), rabbit anti-BLM (A300-110, Bethyl Labs; dilution of 1:1,000), and rabbit anti-53BP1 (A300-272A, Bethyl Labs; dilution of 1:1,000). Cells were washed with PBS containing 1% FBS and stained for 2 h in the dark at room temperature with Alexa Fluor 594 goat anti-rabbit (eBioscience) and Alexa Fluor 488 goat anti-mouse (eBioscience) at dilutions of 1:1,000. Slides were prepared with mounting media with 4,6-diamidino-2-phenylindole (DAPI; Vectashield). Confocal immunofluorescence images were collected at 405, 488 and 561 nm using a Yokogawa CSU-X1 spinning disk confocal (Andor Technology) mounted on a Nikon Ti-E inverted microscope (Nikon Insruments, Melville, NY). Images were acquired using a $\times 40$ Plan Apo NA 1.0 oil objective with an Andor iXon 897 EMCCD camera. Acquisition parameters, shutters and filter positions were controlled using the Andor iQ software. Confocal microscopy images for this study were acquired in the Confocal and Light Microscopy Core facility at the Dana-Farber Cancer Institute. Additional images were acquired using a Zeiss AxioImager Z1 microscope equipped with an Axiocam MRc Rev.3 colour digital camera and Plan APO $\times 63/1.4$ oil M27 lens (magnification $\times 63$). Acquisition software and image processing used the Zeiss AxioVision software package (Zeiss Imaging). Images were analysed with the US National Institutes of Health Image J FIJI program (http://rsbweb.nih.gov.ezp-prod1.hul.harvard.edu/ij/index.html). Specifically, ImageJ FIJI[71] was used to quantify foci in immunofluorescence experiments. Images from the vehicle-treated condition were used to set an appropriate noise tolerance for each antibody stain. The same noise threshold was used for all images in a given experiment (noise tolerances were as follows: 10 for γH2AX (JBW301), 10 for G4 DNA (1H6) and 5 for 53BP1 (Bethyl)). The 'Find Maxima' tool was used to count foci for the indicated number of nuclei. Just another co-localization plugin (Jacop)[72] was used to calculate degree of co-localization between foci.

**Colony survival assay.** Cells were seeded at a density of 200 cells per 10 cm tissue culture plate and 24 h later treated with the indicated dose of genotoxin. Cells were treated with bleomycin for 1 h and pyridostatin for 48 h before washing twice with PBS, adding fresh media and returning to the incubator for 10 days to develop colonies. Colonies were fixed with methanol and stained with 0.5% crystal violet for counting. Colony survival assays are presented as representative experiments from $n = 3$ biological replicates.

**Propidium iodide cell cycle assay.** Cells were irradiated and pulsed with 10 µM BrdU (Sigma) for 1 h at the indicated time points and fixed in 35% ethanol in DMEM. Cells were resuspended and incubated in 2 N HCl for 20 min, washed in 0.1 M sodium borate, pH 8.5, and washed in PBS. Subsequently, cells were stained with anti-BrdU FITC-conjugated antibody (Clone B44, BD Bioscience) in a solution of 0.5% Tween-20 and 0.5% bovine serum albumin. Cells were stained with propidium iodide in PBS containing RNase before analysis by flow cytometry. For propidium iodide analysis alone, cells were fixed in 35% ethanol and stained with propidium iodide before being run on a FACS Canto II (BD Bioscience). The data were analysed using FlowJo to determine the cell cycle distribution. All flow cytometry in this study was performed at the Hematologic Neoplasia Flow Cytometry Core at Dana-Farber Cancer Institute.

**DR-GFP and EJ5 assays.** U2OS cells containing the DR-GFP assay targeted to the *AAVS1* locus have already been described[73]. The EJ5 reporter[55] was cloned and randomly integrated in HeLa cells with puromycin selection. Cells were transfected with siCont or siPARP3 siRNA pools (Dharmacon) and, 24 h following transfection, were infected with Ad I-SceI at $2 \times 10^9$ p.f.u. ml$^{-1}$. Forty-eight hours following infection, cells were analysed using flow cytometry.

**Chromatin immunoprecipitation.** ChIP assays were performed using the SimpleChIP Chromatin IP kit (Cell Signaling Technology). Cells were crosslinked with formaldehyde (5 min) and quenched with glycine. Cell pellets were solubilized in ChIP buffer (Cell Signaling Technology) and sonicated (Fisher BioRuptor U200). Part of the supernatant was digested with proteinase K (65 °C for 2 h), the DNA isolated by spin columns, and input DNA quantified by real-time PCR. Equivalent amounts of chromatin were incubated with primary antibody (overnight at 4 °C for RPA, CtIP, and BLM and HA, 2 h at 4 °C for γH2AX) followed by protein-G magnetic beads (Cell Signaling Technology). Immune complexes were washed in low- and high-salt ChIP buffers (Cell Signaling Technology), eluted and incubated in NaCl (65 °C for 2 h), and then digested with proteinase K. Purified DNA was quantified by qPCR using the Step One Plus real time PCR system (Applied Biosystems). ChIP-grade antibodies included anti-HA magnetic beads (HA-7, Pierce), mouse anti-RPA34 (NA19L, Calbiochem), mouse anti-γH2AX (JBW301, Millipore), rabbit anti-CtIP (A300-487, Bethyl Labs), rabbit anti-BLM (A300-110, Bethyl Labs) and normal mouse IgG (Santa Cruz). The anti-RPA34 (NA19L, Calbiochem) and mouse anti-γH2AX (JBW301, Millipore) reagents have been used extensively for ChIP[74,75]. Primer sequences are listed in Supplementary Table 5.

**Preparation of hf2 antibody.** pSANG10-3F-BG4 (Addgene 55756)[52] was transformed into BL21(DE3) competent *Escherichia coli*, (Invitrogen) grown in LB broth and induced for 6 h with 10 mM isopropyl-β-D-thiogalactoside (Life Technologies). Bacterial pellets were snap-frozen, lysed and sonicated (Fisher BioRuptor U200) before centrifuging at 12 K r.p.m. Lysates were incubated at 4 °C with the end-over-end rotation overnight with Flag-agarose beads (Sigma Aldrich). Beads were washed in lysis buffer (50 mM Tris, pH 7.5, 150 mM NaCl, 0.05% NP-40) and used immediately for IP go G4 DNA.

**Preparation of DNA and IP of G4 DNA with hf2 antibody.** Wild-type or *PARP3*$^{-/-}$ 293T cells were treated with 5 μM pyridostatin for 24 h and transfected with the pX330 plasmid expressing CAS9 with or without a gRNA targeting the *CD4* locus 16 h before isolation of DNA. Isolation of genomic DNA and pulldown of G4 DNA were performed as described[52] with slight modifications. Genomic DNA was extracted from 293T cells using the QIAamp DNA Mini kit (Qiagen) according to the manufacturer's protocol. The purified DNA was eluted from columns in 10 mM Tris, pH 7.4. Genomic DNA was sheared by sonication (Fisher BioRuptor U200) before pull-down experiment. An amount of 60 μg of DNA was used per condition and incubated overnight with Flag beads bound to hf2. Following 4 °C incubation overnight with the end-over-end rotation, beads were washed six times with 10 mM Tris, pH 7.4, 100 mM KCl and 0.1% Tween followed by one wash with 10 mM Tris pH 7.4, 100 mM KCl. DNA was eluted in 50 μl of 1% SDS, 0.1 M NaHCO$_3$ at 30 °C for 1 h and then purified with Qiagen PCR purification columns. Purified DNA was quantified by qPCR using the Step One Plus real time PCR system (Applied Biosystems). Primer sequences are listed in Supplementary Table 7.

**Statistics.** Statistical tests used were unpaired, two-sided Student's *t*-test to test for differences between the means of two independent sets of normally distributed data. $P < 0.05$ as the significance level was used. No marked differences in variance within individual experiments were observed across conditions. Error bars show s.e.m. On the basis of the observations from previous shRNA screens[76], we stipulated that each replicate included ≥200,000 transgene-positive cells to ensure that each shRNA in the library was represented by ≥200 cells. No statistical method was used to predetermine sample size in this or other experiments.

**Data availability.** DNA primer sequences and CRISPR-Cas9 gRNA sequences are all found in Supplementary Tables 5–7. Any other relevant sequences are available upon request from the authors. RNA-seq data are publicly available in the NCBI GEO repository under the accession number GSE94588.

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

## Acknowledgements

We thank Dipanjan Chowdhury, Cyrus Vaziri, Andrew Lane and members of the Weinstock and Price laboratories for thoughtful comments. This work was supported by American Cancer Society and Alex Lemonade Stand fellowships (both to T.A.D.), Claudia Adams Barr Fund (to T.A.D. and D.M.W.), and NIH/NCI R01 CA151898 and R01 CA172387 (both to D.M.W.).

## Author contributions

T.A.D., A.C.S., J.D., D.E.R., W.C.H. and D.M.W. conceived and designed the rearrangement factor screen; T.A.D., J.V.L., B.D.P. and D.M.W. conceived and designed the experiments; T.A.D., J.V.L., J.P.C., S.G., T.T., S.K. and F.I. performed the experiments, T.A.D., K.E.S., A.C.S., B.D.P. and D.M.W. analysed and interpreted the data; T.A.D. and D.M.W. wrote the paper; and J.D. and B.D.P. contributed to preparing the manuscript.

## Additional information

**Competing interests:** The authors declare no competing financial interests.

DOI: 10.1038/ncomms15918    OPEN

# Corrigendum: PARP3 is a promoter of chromosomal rearrangements and limits G4 DNA

Tovah A. Day, Jacob V. Layer, J. Patrick Cleary, Srijoy Guha, Kristen E. Stevenson, Trevor Tivey, Sunhee Kim, Anna C. Schinzel, Francesca Izzo, John Doench, David E. Root, William C. Hahn, Brendan D. Price & David M. Weinstock

Nature Communications 8:15110 doi: 10.1038/ncomms15110 (2017); Published 27 Apr 2017; Updated 13 Jun 2017

In this Article, the antibody 'BG4' is consistently referred to incorrectly as 'hf2'. These errors appear in the Results and Methods. Additionally, the Article incorrectly cites reference 52 in the sentences 'A recent study described hf2, an engineered single-chain antibody specific for G4 DNA[52]' and 'pSANG10-3F-BG4 (Addgene 55756)[52] was transformed into….' The correct citation is given below.

Biffi, G., Tannahill, D., McCafferty, J. & Balasubramanian, S. Quantitative visualization of DNA G-quadruplex structures in human cells. *Nat. Chem.* **5**, 182–186 (2013).

