## [Peer Review File · Nature Communications]

Reviewers' Comments:

Reviewer #1 (Remarks to the Author)

This manuscript by Weinstock and collaborators identifies PARP3 as one of the factors that can influence translocation efficiency. They propose that the mechanism somehow involves formation of G-quadruplex structures that inhibit repair of DSBs by homologous recombination or NHEJ. The manuscript shows that inhibition of PARP3 stimulates translocation frequencies about 2-fold in the reporter system and cell lines tested, and those results are persuasive although the effect is not large. However, the experiments presented to support the proposal for G4 DNA involvement are not persuasive. Without some sense of mechanism, the evidence that PARP3 prevents translocation is interesting, but unlikely to have high impact.

Identification of a role for PARP3 in preventing translocation derives from a nice assay, developed by these authors, in which a segment from just downstream of the AAVS1 locus promoter is fused to a downstream CD4 or GFP gene is randomly integrated in the genome of A-549 lung adenocarcinoma cells. Simultaneous targeted cleavage of the AAVS1 segment in the genome and in the reporter by an engineered zinc finger nuclease initiates translocation, which results in expression of the reporter at a frequency of around 1%. This assay was used to screen a library of 169 candidate shRNAs that target DNA repair-associated genes, identifying five candidates that affected translocation frequency in the range of 2-fold: UBC9, RAD50, 53BP1, DDB1 and PARP3. UBC9 and RAD50 were previously identified in a screen for factors involved in translocation in yeast; and 53BP1 is a very plausible candidate based on its functions in Ig switch recombination. All five hits were validated in HeLa cells, good evidence that the assay is robust. Altogether, identification of this path

PARP3 is not essential and it is a druggable target, rationalizing further study focused on its mechanism of action. Evidence that the G4 ligand, pyridostatin, is especially toxic to PARP3^{-/-} cells (Fig. 4) leads to the proposal that G4 DNA might be involved in the mechanism of PARP3 function in translocation. Pyridostatin has been shown by Steve Jackson's lab to induce DNA damage throughout the genome (Rodriguez et al, 2012, PMID 22306580). However, that damage does not cleanly map to G4 motifs, nor does any G4 ligand cleanly cause damage at G4 motifs in human cells.

The major caveat in interpreting the synergy between PARP3 knockdown and pyridostatin as affecting G-quadruplexes is that both treatments are pleiotropic. PARP3 was recently shown to affect transcription (Gibson et al., 2016, PMID 27256882), as do G4 ligands (e.g. Halder et al. 2012 PMID 22414013). One explanation for observed effects is transcriptional deregulation of one or more genes upon these combined treatments.

Instead, the manuscript attempts to provide evidence that supports a model in which PARP3 normally counteracts G-quadruplex formation at the ends of DSBs that could be sites of translocation. The experiments shown in the last few figures are not strong. For example:

- Events are compared at only two DSB sites, in genes with higher and lower concentrations of G4 motifs, CD4 (one of the two reporters) and ESR1 (which encodes the estrogen receptor). For the results to be convincing, DSBs must be targeted to other regions.
- BLM helicase is tested as a key candidate for regulation of quadruplex structures, without clearly rationalizing that choice.
- ChIP data are presented claiming that increased occupancy of BLM at DSBs in CD4 relative to ESR1 is evidence of involvement of G-quadruplex structures. In fact, there is no compelling evidence in the literature for a requirement for BLM in maintaining genomic stability at G-quadruplexes, so if BLM were enriched at DSBs in CD4 it would be difficult to interpret the meaning of that result.

- The ChIP data are not particularly strong, and the claimed involvement makes one wonder whether, if BLM is this important, it shouldn't have been identified as a candidate in the initial the screen. Was it tested? How did it score?
- The manuscript lacks controls demonstrating specificity of the anti-BLM antibody (or other antibodies) used for ChIP,

Other comments:

The manuscript does not present specificity controls for the antibodies used in ChIP analyses.

The hf2 and IH6 antibodies bind G4 DNA in vitro, but neither has been shown to enrich sequences that form G4 motifs when used for ChIP with genomic DNA, in prior publications or this one. Thus neither represents an appropriate assay for G4 DNA in vivo.

Fig. 2c, assay of 53BP1: There is overexpression not knockdown in lane 3 judging by the gel below (not well aligned with the graph) and the increase rather than decrease evident in the translocation frequency. Why was this included here? Is it fair to include this in the R2 calculation, which would lead to an overestimation of the slope and make the effect of knockdown look more significant than it may be?

Fig. 3: shows that PARP3 knockdown stimulates translocation in several different types of human cells (not really multiple as stated, e.g. in Fig. 3 title. (Hela, A549, 293T). What is the p53 status of these cells? Panel 3a needs clearer labeling. There is no panel 3b, and two panels are labeled 3c.

Fig. 4h and j: The two BLM occupancy curves look very similar to me, even though the authors claim only the occupancy at the ESR is significant.

Fig. 5 - PARP3 protects against damage and pyridostatin causes it, but both act pleiotropically and by pathways that are not completely defined. Thus it is not surprising or really relevant that the combination of these treatments causes

Fig. 6: The authors have tested for enrichment of repair factors only at CD4, although for ESR1 to be included in Fig. 4 it should also be shown to respond to cleavage by accumulation of CtIP, RPA and gH2AS.

Reviewer #2 (Remarks to the Author)

Day et al.

PARP3 is a promoter of chromosomal rearrangements and limits G4 DNA

This manuscript reports a siRNA screen for modulators of the frequency of chromosomal translocation induced by zinc finger nuclease-mediated double strand breaks. In the first part of the paper, the authors report the identification of five 'hits' of which they choose to focus on one, PARP3. Knockdown of PARP3 reduces the frequency of translocations. The second part of the manuscript attempts to provide a mechanistic explanation for the role of PARP3. Based on the observation that PARP3-deficient cells are hypersensitive to the drug pyridostatin, which stabilises G quadruplex (G4) DNA secondary structures, the authors develop a model in which the absence of PARP3 leads to increased G4 formation and inhibition of resection, explaining the defect in translocation induction.

There are some potentially interesting observations reported in this paper. However, I am not convinced that the conclusions, particularly with respect to the relationship between PARP3 and G4s in resection, are supported by the presented data. Many of the results are incompletely controlled and, in places, over interpreted.

Specific points:

1. The screen. I do not find the results from the screen particularly compelling as it is unclear why only five genes are identified. Several expected 'hits' are not seen, for instance ligase IV and XRCC4 (as reported in Ghezraoui et al. 2014) or, given the identification of RAD50, MRE11 and NBS1. Further, several aspects of the screen are not well explained:

- What are the absolute numbers of GFP positive cells identified in the different lines? This is helpful in interpreting the robustness of the fold-changes reported
- Is there any 'background' GFP positivity even in the absence of the zinc finger nuclease expression.
- Do the authors know that every GFP positive event corresponds to a translocation to the AAVS1 locus? Since the GFP or CD4 construct is randomly integrated into the genome it is possible that the upstream cut could also result in deletions that in some circumstances could bring the reporter under the influence of another promoter close to the integration site
- For each line, do the authors know how many copies of the GFP or CD4 reporters are integrated?
- Do the three GFP and four CD4-reporter repeats represent different integrations or just repeats on the same integration? This is important as the context of the integration could affect translocation frequency and / or the dependence on particular proteins. It seems that the correlation between the GFP and the CD4 reporters is actually quite poor (Figure 1E). r looks to be low on these plots.

2. Why is no Western blot show for PARP3 in Figure 2?

3. Figure 3 presents a lot of information derived from different cell lines, but

4. The hypersensitivity of PARP3ko cells to pyridostatin is interesting. However, this key result needs to be made more robust and convincing. Is the effect complemented by ectopic expression of PARP3 and is it seen with other G4 stabilising ligands, for instance NMM or PhenDC3, both of which are commercially available?

5. The experiments in which various components of the DSB response are ChIP'd from around the break site in the CD4 locus are interpreted as showing an influence of G4s in PARP3-deficient cells. However, this is an over interpretation of a correlation. BLM occupancy, for instance (Figure 4h & j) is increased around the break. However, this does not necessarily have anything to do with G4s. BLM does indeed play a role in processing G4s, but it also plays a role in double strand break repair through Holliday junction dissolution, among other functions. In both the case of CD4 and ESR1, BLM occupancy is associated with a break and may not be related to the presence of G4s. For instance, is its recruitment modulated by exposure to pyridostatin?

6. The G4 antibody immunoprecipitations are also incomplete. The pyridostatin-induced enrichment at the telomere and possibly at the MYC locus is clear. However, Figure 4k does not include the CD4 locus, which would be expected to give a significantly greater signal than ESR1. Does it? Likewise, it would be predicted that if the experiment shown in Figure 4i was repeated at the ESR1 locus that there would not be enrichment of the G4 antibody signal after inducing a DSB. The proposed model would also predict that inhibition of PARP3 (using ME0328, as show in Figure 4f) would have the same effect, while PDS would increase the effect.

7. In Figure 5, the quantitation of foci is highly problematic. Given the picture of gammaH2Ax foci show in 5a, the precision and error shown in the quantitation in Figure 5b seems remarkable. Even so, there is only a significant difference at one time point. The G4 antibody foci are even more worrying as no examples of the IF pictures are shown, and again there is basically no significant effect overall.

8. The conclusion that 'resection is suppressed' is not supported by the data presented in Figure 6. The ChIP experiments show only patchy significance and, I'm afraid, are not convincing.

Reviewer #3 (Remarks to the Author)

In the manuscript "PARP3 is a promotor of chromosomal rearrangement and limits G4 DNA" by Day et al. the authors identified two factors, i.e. UBC9 and RAD50, as suppressors of chromosomal rearrangement, and three factors, i.e., 53BP1, DDB1, and PARP3, as promotors of chromosomal rearrangement in two different human cell lines. In the following they studied the role of PARP3 in these processes in greater detail. They convincingly demonstrate that PARP3 is associated with the regulation of G4 DNA in response to DNA damage. Specifically, they showed that PARP3 KO cells are sensitive to pyridostatin-induced DNA double strand breaks. Furthermore, the authors show that PARP3 supports the processing of DSBs into single-stranded DNA and cooperates with BLM helicase in NHEJ repair.

The study is comprehensive, for the most part the data is of high quality, the methods section is detailed, and the manuscript is well written. The conclusions drawn by the authors and the model developed are supported by their results. In principal, I support publication of this manuscript after considering the following issues:

Fig 2: Could the authors indicate, why they marked some siRNAs by a star? The linear regression analyses should be extended by a statement on the P values.

Fig. 3: Mislabeling of panel b (c∅b). Scale bars are missing in microscopic pictures of panel j. Size of images in panel j should be increased and image quality should be improved. Unit is missing at the x-axis of panel p.

Fig. 4: Panel e: Since a genetic knock-out cannot directly be compared to siRNA-mediated knock-down, PARP3 knock-down should be included in the experiment shown in panel e. Furthermore, the PARP2 Western blot is not convincing. There seems to be a prominent band cut right at the lower edge. In panels h, j, and l, units at the x-axis are missing.

Fig. 5: Evaluation of 53BP1 foci and their degree of colocalization with γH2A.X foci should be analyzed. Scale bars missing in panel a. Quality of images in panel a is not satisfactory. Color code between panels b, c, and d is inconsistent. Choose different color for wild-type (?). Can the authors explain the decrease in G4 DNA after 6 hours in "PARP3-/-, ad b-gal" cells. In panel f, it is not clear which groups have been compared for statistical analysis. 2-Way ANOVA testing seems to be the appropriate statistical test here.

Fig. 6: Units at x-axis are missing in panels b,c,and d. Scale bars are missing in panel e. Can the image quality in panel e be increased?

Fig. 7. In panels a and b the x-axes could be presented as continuous data (instead of categorical data). The extent of colocalization of 53BP1 and γ-H2A.X foci should be analyzed. Panel d could be presented as a separate Figure.

Point-by-point response to Referee's Comments for Manuscript NCOMMS-16-18904-T

Dear Referees,

Thank you for your thoughtful consideration of our manuscript. Please find the point-by-point responses to your comments below in blue text.

Reviewer #1 (Remarks to the Author):

This manuscript by Weinstock and collaborators identifies PARP3 as one of the factors that can influence translocation efficiency. They propose that the mechanism somehow involves formation of G-quadruplex structures that inhibit repair of DSBs by homologous recombination or NHEJ. The manuscript shows that inhibition of PARP3 stimulates translocation frequencies about 2-fold in the reporter system and cell lines tested, and those results are persuasive although the effect is not large. However, the experiments presented to support the proposal for G4 DNA involvement are not persuasive. Without some sense of mechanism, the evidence that PARP3 prevents translocation is interesting, but unlikely to have high impact.

We thank Reviewer #1 for this thorough analysis. As outlined below, we have added extensive new data that further supports the involvement of PARP3 in resolving G4 DNA at break ends.

Identification of a role for PARP3 in preventing translocation derives from a nice assay, developed by these authors, in which a segment from just downstream of the AAVS1 locus promoter is fused to a downstream CD4 or GFP gene is randomly integrated in the genome of A-549 lung adenocarcinoma cells. Simultaneous targeted cleavage of the AAVS1 segment in the genome and in the reporter by an engineered zinc finger nuclease initiates translocation, which results in expression of the reporter at a frequency of around 1%. This assay was used to screen a library of 169 candidate shRNAs that target DNA repair-associated genes, identifying five candidates that affected translocation frequency in the range of 2-fold: UBC9, RAD50, 53BP1, DDB1 and PARP3. UBC9 and RAD50 were previously identified in a screen for factors involved in translocation in yeast; and 53BP1 is a very plausible candidate based on its functions in Ig switch recombination. All five hits were validated in HeLa cells, good evidence that the assay is robust. Altogether, identification of this path [*sic*]

To clarify, our manuscript presents evidence that PARP3 *promotes* translocations, as knockdown of PARP3 reduces translocations. In addition, we used our assay to screen 169 candidate *genes* using ~850 shRNA (including control shRNA), not 169 *shRNA*. The final sentence of the Reviewer's comment was truncated when forwarded to us.

PARP3 is not essential and it is a druggable target, rationalizing further study focused on its mechanism of action. Evidence that the G4 ligand, pyridostatin, is especially toxic to PARP3^{-/-} cells (Fig. 4) leads to the proposal that G4 DNA might be involved in the mechanism of PARP3 function in translocation. Pyridostatin has been shown by Steve

Jackson's lab to induce DNA damage throughout the genome (Rodriguez et al, 2012, PMID 22306580). However, that damage does not cleanly map to G4 motifs, nor does any G4 ligand cleanly cause damage at G4 motifs in human cells.

We share the Reviewer's concern that the DNA damage caused by pyridostatin in Rodriguez *et al.*, 2012¹ does not align perfectly with G4 motifs. However, we note that the authors find a clear enrichment for pyridostatin-induced DNA damage (as assayed by ChIP-seq for γ H2AX) in genes containing high 'putative G quadruplex-forming sequences' (shown in Figure 5c of that manuscript).

Furthermore, results in the recent Chambers *et al.*, 2015² manuscript support the notion that pyridostatin preferentially binds and stabilizes regions of G4 DNA. The authors used the observation that "polymerase stalling at G4 sites...affect[s] base calling," and leads to mismatches in the sequence. After pyridostatin treatment, there is a significantly higher mismatch level in predicted quadruplex sequences (~35%) than elsewhere (<2%) when compared across the genome (Figure 2b). Based on these results, we believe there is sufficient evidence to employ pyridostatin as a tool for preferentially targeting damage to regions of high G4 DNA content.

The major caveat in interpreting the synergy between PARP3 knockdown and pyridostatin as affecting G-quadruplexes is that both treatments are pleiotropic. PARP3 was recently shown to affect transcription (Gibson et al., 2016, PMID 27256882), as do G4 ligands (e.g. Halder et al. 2012 PMID 22414013). One explanation for observed effects is transcriptional deregulation of one or more genes upon these combined treatments.

We note that while Gibson *et al.*, 2016 demonstrates that PARP1 affects transcription (Figure S16 and Table S2), the paper does not present direct evidence that PARP3 affects transcription. Nonetheless, we have significantly expanded the scope of the manuscript by performing RNA-sequencing (RNA-seq) to define the transcriptional consequences of PARP3 loss. Specifically, we performed RNA-seq (in triplicate) on wild-type and *PARP3*^{-/-} A549 cells with and without pyridostatin treatment. We chose these cells and this dose of pyridostatin (1.0 μ M), as these were the conditions under which we observed PARP3-dependent sensitivity in **Figure 4d**.

In the absence of pyridostatin, only 61 genes were significantly differentially expressed between wild-type and *PARP3*^{-/-} cells with an adjusted *p*-value ≤ 0.05 and the absolute value of the log₂ fold-change at ≥ 1 . The top 10% of differentially expressed genes (n=6) are listed in **Table A** below. All 61 genes are reported in **Supplementary Table 3** and a heat-map showing clustering of replicates in all 61 significantly differentially expressed genes is included as **Supplementary Figure 5**.

Table A – Six most highly differentially expressed genes between untreated wild-type and untreated *PARP3*^{-/-} A549 cells.

Gene	log ₂	Gene Name	Annotation
------	------------------	-----------	------------

Symbol	fold-change		
CPS1	-2.870670458	Carbamoyl-Phosphate Synthase 1	A mitochondrial enzyme that catalyzes synthesis of carbamoyl phosphate from ammonia and bicarbonate, the first step in the urea cycle.
PXDN	2.538435519	Peroxidasin	A heme-containing extracellular matrix peroxidase that is involved in extracellular matrix formation
PTPRD	-2.544686444	Protein Tyrosine Phosphatase, Receptor Type D	A transmembrane protein tyrosine phosphatase
H19	-2.245936667	H19, Imprinted Maternally Expressed Transcript	A long non-coding RNA which functions as a tumor suppressor.
CYP4F11	-2.474712512	Cytochrome P450 family 4 subfamily F member 11	A member of the cytochrome P450 superfamily of monooxygenases which catalyze many reactions involved in drug metabolism and synthesis of cholesterol, steroids and other lipids.
UPK1B	2.06610896	Uroplakin 1B	Component of the asymmetric unit membrane, a highly specialized biomembrane elaborated by terminally differentiated urothelial cells.

Using the 61 significantly differentially expressed genes between wild-type and *PARP3*^{-/-} cells, we performed Gene Ontology (GO) Term Enrichment analysis (**Supplementary Figure 6a**). Of the top 10 enriched GO terms, we found multiple related to development but none suggesting significant alterations in DNA repair pathways. This is included as the new **Supplementary Figure 6a** and below. Indeed, we did not identify any enriched GO terms related to response to DNA damage in any of the 22 enriched GO terms that would suggest pleiotropic transcriptional effects. These points are now clearly outlined in the manuscript text.

Supplementary Figure 6a – GO term enrichment in 61 genes significantly differentially expressed genes between wild-type and *PARP3*^{-/-} A549 cells with vehicle treatment.

To investigate pathway enrichment using an orthogonal tool, we performed Kyoto Encyclopedia of Genes and Genomes (KEGG) pathway analysis using the DAVID Bioinformatics database^{3,4}. The genes that were differentially expressed between wild-type and *PARP3*^{-/-} cells in the vehicle treatment were found to be significantly enriched in the KEGG pathways of 1) complement and coagulation cascade, 2) hematopoietic cell lineage, and 3) platelet activation (**Supplementary Figure 6b** shown below). Considering that A549 cells are lung adenocarcinoma cells, these pathway alterations are likely irrelevant, suggesting that the transcriptional effects from PARP3 loss are minimal.

Supplementary Figure 6b – KEGG pathway analysis showing pathways that were significantly enriched in the differentially expressed genes between wild-type and *PARP3*^{-/-} cells with either vehicle treatment (top) or pyridostatin (PDS) treatment (bottom).

Next, we examined transcriptional differences between wild-type and *PARP3*^{-/-} A549 cells following pyridostatin treatment. We found that 64 genes were significantly differentially expressed between wild-type and *PARP3*^{-/-} cells following pyridostatin treatment with a *p*-value adjusted to 0.05 and the absolute value of the log₂ fold-change at > 1. Again, the most highly differentially expressed genes are listed in **Table B** below (with a list of all 64 genes reported in **Supplementary Table 4**) and a heat-map showing clustering of replicates in all 64 significantly differentially expressed genes is shown in **Supplementary Figure 7**.

Table B – Most highly differentially expressed genes between pyridostatin-treated wild-type and pyridostatin-treated *PARP3*^{-/-} A549 cells. Of the 6 genes in **Table A**, 5 are included among the top 7 below.

Gene Symbol	log ₂ fold-change	Gene Name	Annotation
CPS1	-3.024545179	Carbamoyl-Phosphate Synthase 1	A mitochondrial enzyme that catalyzes synthesis of carbamoyl phosphate from ammonia and bicarbonate, the first step in the urea cycle.
PXDN	2.534107973	Peroxidasin	A heme-containing extracellular matrix peroxidase that is involved in extracellular matrix formation
RRAD	2.165775082	Ras Related Glycolysis Inhibitor And Calcium Channel Regulator	May play an important role in cardiac antiarrhythmia via the strong suppression of voltage-gated L-type Ca(2+) currents.
PTPRD	-2.788062107	Protein Tyrosine Phosphatase, Receptor	A transmembrane protein tyrosine phosphatase

		Type D	
CYP4F11	-2.535549093	Cytochrome P450 family 4 subfamily F member 11	A member of the cytochrome P450 superfamily of monooxygenases which catalyze many reactions involved in drug metabolism and synthesis of cholesterol, steroids and other lipids.
H19	-2.224701699	H19, Imprinted Maternally Expressed Transcript	A long non-coding RNA which functions as a tumor suppressor.
UPK1B	2.00321143	Uroplakin 1B	Component of the asymmetric unit membrane, a highly specialized biomembrane elaborated by terminally differentiated urothelial cells.

We noted that the differentially expressed genes in the vehicle-treated and pyridostatin-treated comparisons of wild-type and *PARP3*^{-/-} cells were >80% identical (Supplementary Tables 3 and 4), suggesting relatively minor differences in the transcriptional response to pyridostatin across genotypes. Table C lists the 11 genes that were differentially expressed between wild-type cells treated with pyridostatin and *PARP3*^{-/-} cells treated with pyridostatin. None of these genes is reported to function in DNA repair nor is any of them implicated more broadly in a pathway governing cell death.

Table C – The 11 genes that are differentially expressed between wild-type and *PARP3*^{-/-} cells *only* upon treatment with pyridostatin.

Gene Symbol	log ₂ fold-change	Gene name	Annotation
CELF2	-1.180805614	CUGBP, Elav-like family member 2	RNA binding protein implicated in mRNA splicing
CPLX2	-1.095622433	Complexin 2	Cytosolic protein functioning in synaptic vesicle exocytosis
PDK4	-1.082592671	Pyruvate Dehydrogenase Kinase 4	A mitochondrial protein that plays a role in glucose and fatty acid metabolism
CEMIP	-1.048034657	Cell Migration Inducing Hyaluronan Binding Protein	Mediates depolymerization of hyaluronic acid by the cell membrane-associated clathrin-coated pit endocytic pathway.
HYAL1	-1.034515953	Hyaluronoglucosaminidase 1	A lysosomal hyaluronidase that degrades hyaluronan, one of the major glycosaminoglycans of the extracellular matrix.
PCSK9	-1.012504769	Proprotein Convertase	A protease that processes proteins

		Subtilisin/Kexin Type 9	passing through the secretory pathway.
NT5E	1.043102796	5'-Nucleotidase Ecto	A plasma membrane protein that catalyzes the conversion of extracellular nucleotides to membrane-permeable nucleosides.
CORO2B	1.052529898	Coronin 2B	May play a role in the reorganization of neuronal actin structure.
CDH6	1.066224679	Cadherin 6	A membrane glycoprotein that mediates homophilic cell-cell adhesion
FEZ1	1.111122026	Fasciculation And Elongation Protein Zeta 1	May be involved in axonal outgrowth as component of the network of molecules that regulate cellular morphology and axon guidance machinery.
MUC3A	1.223058975	Mucin 3A, Cell Surface Associated	An epithelial glycoprotein that is thought to be a component of a variety of mucus gels.

Using the 64 significantly differentially expressed genes between wild-type and *PARP3*^{-/-} cells following pyridostatin treatment, we again performed GO Term Enrichment analysis (**Supplementary Figure 8** shown below). Again, none of the 24 significantly enriched GO terms included any related to DNA repair or pathways that might simply explain the dramatic pyridostatin sensitivity observed in *PARP3*^{-/-} cells.

performed KEGG pathway analysis on the genes that were differentially expressed following pyridostatin treatment. This analysis yielded an almost identical result to the

pathway analysis of the vehicle-treated cells, which is not surprising given the similarity of the differentially expressed genes between the two sets. The finding that neither PARP3-deficiency alone nor PARP3-deficiency combined with pyridostatin treatment leads to transcriptional changes that would explain the acute sensitivity to pyridostatin observed in PARP3-deficient cells suggests that transcriptional deregulation does not underlie the DNA repair phenotypes outlined in our manuscript. We have added this analysis to the manuscript, which we believe significantly enhances the contribution to the literature.

Instead, the manuscript attempts to provide evidence that supports a model in which PARP3 normally counteracts G-quadruplex formation at the ends of DSBs that could be sites of translocation. The experiments shown in the last few figures are not strong. For example:

— Events are compared at only two DSB sites, in genes with higher and lower concentrations of G4 motifs, CD4 (one of the two reporters) and ESR1 (which encodes the estrogen receptor).

For the results to be convincing, DSBs must be targeted to other regions.

We thank the Reviewer for this suggestion and have included 6 additional sites of targeted damage with higher or lower concentrations of G4 motifs. Please see new **Supplementary Figure 15** (excerpted below). This data confirms the finding that

PARP3 loss preferentially affects RPA accumulation at DSB ends formed at loci with significant G4 DNA.

chr19:55,626,140-55,628,139 (38.9% G4 DNA)

chr7:66,613,994-66,615,993 (0% G4 DNA)

chr1:226,594,594-226,596,593 (17.6% G4 DNA)

chr3:195,806,437-195,808,436 (5.6% G4 DNA)

chr15:43,783,500-43,785,499 (19.4% G4 DNA)

chrX:70,508,601-70,510,600 (0% G4 DNA)

— BLM helicase is tested as a key candidate for regulation of quadruplex structures, without clearly rationalizing that choice.

We apologize that this choice was not clearly rationalized and we have added explicit discussion of the relationship between BLM and G4 DNA with references (excerpted below):

“The BLM helicase is known to bind and unwind G4 DNA *in vitro*⁵⁻⁸ and can serve as a surrogate to map G4 DNA locations *in vivo*^{9,10}. In addition, BLM facilitates replication through G4 DNA in telomeric sequences *in vivo*¹¹ and BLM-deficient cells exhibit G4 DNA motif-associated epigenetic instability¹². Taken together, these data suggest that G4 DNA is a physiological substrate for BLM helicase that accumulates at DNA damage targeted to regions with abundant G4 DNA.”

— CHIP data are presented claiming that increased occupancy of BLM at DSBs in CD4 relative to ESR1 is evidence of involvement of G-quadruplex structures. In fact, there is no compelling evidence in the literature for a requirement for BLM in maintaining genomic stability at G-quadruplexes, so if BLM were enriched at DSBs in CD4 it would be difficult to interpret the meaning of that result.

Please see the discussion of the previous point.

— The CHIP data are not particularly strong, and the claimed involvement makes one wonder whether, if BLM is this important, it shouldn't have been identified as a candidate in the initial screen. Was it tested? How did it score?

Indeed, BLM was included in the original DNA repair library (Supplementary Table 1). However, none of the 5 shRNA targeting BLM resulted in significant changes in the frequency of chromosomal rearrangements (see below). The Reviewer brings up a much larger point about false-negative findings in shRNA screens, which can result from a number of factors, including: 1) toxicity of individual shRNA, 2) incomplete knockdown by shRNA, and 3) off-target effects. In addition, it is possible that BLM serves other roles in addition to its role at G4 DNA that counterbalance any effect on translocations and together result in no nonsignificant changes in translocation frequency from BLM knockdown.

— The manuscript lacks controls demonstrating specificity of the anti-BLM antibody (or other antibodies) used for ChIP,

We thank the Reviewer for alerting us to the omission of these important controls. The revised manuscript includes specificity controls for antibodies that have not been used extensively for ChIP (Please see new panels **Supplementary Figure 9b,c**). In addition, we have added references for those antibodies that are routinely used for ChIP and have been validated elsewhere.

Other comments:

The manuscript does not present specificity controls for the antibodies used in ChIP analyses.

Please see above comment and new panels in revised **Supplementary Figure 9b,c**.

The hf2 and IH6 antibodies bind G4 DNA *in vitro*, but neither has been shown to enrich sequences that form G4 motifs when used for ChIP with genomic DNA, in prior publications or this one. Thus neither represents an appropriate assay for G4 DNA *in vivo*.

We share the Reviewer's concern that these reagents to detect G4 DNA are relatively new and have not been used extensively. We note that in our manuscript we have used hf2 to bind G4 DNA *in vitro* in a manner nearly identical to what was published in Lam *et al.*, *Nature Communications* 2013¹³. However, to address the concern that hf2 is a relatively new reagent, we performed multiple control IPs using loci with well-established high G4 DNA content (telomeric sequence, *MYC*) and low G4 DNA content (*ESR1*) (**Figure 4k**). To further address the Reviewer's comment, we have performed and added 2 additional controls for the hf2 IP in **Figure 4k** with the corresponding G4 DNA content for these loci in **Supplementary Figure 9g,h**.

We want to emphasize that we have only used the 1H6 antibody for immunofluorescence (IF) studies (**Figure 5c**) and not for ChIP. This antibody has been used in the context of IF in several studies, including references¹⁴⁻¹⁷. Furthermore, we performed additional controls to ensure that we could increase the 1H6 IF signal in our system using

two small molecule stabilizers of G4 DNA (shown to the right, ** $p < 0.01$; *** $p < 0.001$).

Fig. 2c, assay of 53BP1: There is overexpression not knockdown in lane 3 judging by the gel below (not well aligned with the graph) and the increase rather than decrease evident in the translocation frequency. Why was this included here? Is it fair to include this in the R² calculation, which would lead to an overestimation of the slope and make the effect of knockdown look more significant than it may be?

We thank the reviewer for pointing out this inconsistency. We have recalculated the R² in the assay of 53BP1 excluding the ‘overexpression’ observed in lane 3 and the revised **Figure 2c** reflects the corrected value ($r^2 = 0.3607$).

Fig. 3: shows that PARP3 knockdown stimulates translocation in several different types of human cells (not really multiple as stated, e.g. in Fig. 3 title. (Hela, A549, 293T). What is the p53 status of these cells? Panel 3a needs clearer labeling. There is no panel 3b, and two panels are labeled 3c.

We have corrected the figure legend in Figure 3 to include the word ‘several’ instead of ‘multiple.’ Here again, we note that PARP3 knockdown *inhibits* instead of ‘stimulates’ translocation frequency. Additional labeling has been added to panel 3a to clarify the figure and the duplicate panels have been corrected. We thank the Reviewer for catching these errors. The p53 status of the reported cell lines is as follows: HeLa cells are p53-deficient¹⁸, A549 cells are p53-proficient¹⁹, and 293T cells are p53-deficient owing to SV40 infection²⁰.

Fig. 4h and j: The two BLM occupancy curves look very similar to me, even though the authors claim only the occupancy at the ESR is significant.

We apologize for the confusion. In fact, we state that there was *no* significant difference between BLM occupancy in wild-type and *PARP3*^{-/-} cells at the *ESR1* locus (**Figure 4j**, none of the *p*-values for ESR1/EV were ≤ 0.05). We have contrasted this with the observation that BLM occupancy is statistically significantly higher in *PARP3*^{-/-} cells than it is in wild-type cells at the *CD4* locus (**Figure 4h**), with differential enrichment at three sites flanking the targeted break ($p \leq 0.05$).

We edited the corresponding text from the manuscript (excerpted below) to enhance clarity:

“...we examined the recruitment of BLM to a CRISPR/CAS9-targeted DSB in the *CD4* locus by ChIP. The ChIP signal given by the BLM antibody was validated by siRNA specificity control shown in **Supplementary Fig. 9b**. BLM recruitment was significantly increased in *PARP3*^{-/-} cells relative to wild-type cells (**Figure 4h**), suggesting a greater accumulation of G4 DNA at sequences flanking the DSB in the absence of PARP3. Consistent with this observation, overall chromatin-bound BLM was increased in the absence of PARP3 (**Supplementary Fig. 9d,e**). In contrast to the *CD4* locus, the estrogen receptor 1

(*ESR1*) locus is predicted and observed to contain relatively little G4 DNA (Figure 4i). We therefore hypothesized that PARP3 status would not influence the recruitment of BLM to a CRISPR/CAS9-targeted DSB at *ESR1*. As predicted, we found there was no significant difference between the recruitment of BLM to a DSB in *ESR1* between wild-type and *PARP3*^{-/-} cells (Figure 4j).”

Fig. 5 - PARP3 protects against damage and pyridostatin causes it, but both act pleiotropically and by pathways that are not completely defined. Thus it is not surprising or really relevant that the combination of these treatments causes [*sic*].

We agree that pleiotropic effects are always a concern when examining any factors (genetic or chemical) that modify DNA structure or damage response. We have attempted to utilize both agnostic (RNA-seq) and targeted assays to address these effects, as outlined above. The final sentence was cutoff in the Reviewer’s comment.

Fig. 6: The authors have tested for enrichment of repair factors only at CD4, although for *ESR1* to be included in Fig. 4 it should also be shown to respond to cleavage by accumulation of CtIP, RPA and γ H2AS.

As a result of this useful suggestion, we have performed ChIP for all DNA repair factors suggested (CtIP, RPA, and γ H2AX) at the *ESR1* locus and included this new data in **Supplementary Figure 14**. We note that in contrast to our findings at the *CD4* locus, we find no significant differences between accumulation of these repair factors at a targeted break in *ESR1* between wild-type and *PARP3*^{-/-} cells. These observations are consistent with our model, as the *ESR1* locus contains relatively little G4 DNA.

Reviewer #2 (Remarks to the Author):

Day et al.

PARP3 is a promoter of chromosomal rearrangements and limits G4 DNA

This manuscript reports a siRNA screen for modulators of the frequency of chromosomal translocation induced by zinc finger nuclease-mediated double strand breaks. In the first part of the paper, the authors report the identification of five 'hits' of which they choose to focus on one, PARP3. Knockdown of PARP3 reduces the frequency of translocations. The second part of the manuscript attempts to provide a mechanistic explanation for the role of PARP3. Based on the observation that PARP3-deficient cells are hypersensitive to the drug pyridostatin, which stabilises G quadruplex (G4) DNA secondary structures, the authors develop a model in which the absence of PARP3 leads to increased G4 formation and inhibition of resection, explaining the defect in translocation induction.

There are some potentially interesting observations reported in this paper. However, I am not convinced that the conclusions, particularly with respect to the relationship between PARP3 and G4s in resection, are supported by the presented data. Many of the results are incompletely controlled and, in places, over interpreted.

We thank the Reviewer for this thoughtful consideration. As outlined below and in the response to Reviewer 1, we have provided extensive new data, including the requested controls, to strengthen the conclusion that PARP3 regulates the resolution of G4 DNA during DSB repair.

Specific points:

1. The screen. I do not find the results from the screen particularly compelling as it is unclear why only five genes are identified. Several expected 'hits' are not seen, for instance ligase IV and XRCC4 (as reported in Ghezraoui et al. 2014) or, given the identification of RAD50, MRE11 and NBS1.

The Reviewer brings up a much larger point about false-negative findings in shRNA screens, which can result from a number of factors, including: 1) toxicity of individual shRNA, 2) incomplete knockdown by shRNA, and 3) off-target effects. Of these, the first point is most salient in our screen. Knockdown of essential genes (e.g. MRE11, KU70) will result in cell death and thus, the effects of shRNA targeting these factors on translocation frequency is very difficult to discern. As we note in the text:

“Of the 169 genes, 5 merited further consideration, based on ≥ 2 shRNA with ≥ 2 -fold change in the same direction compared to controls, without causing significant toxicity (Figure 1e, Supplementary Fig. 2a, Supplementary Table 2). It is likely that additional factors assayed in the screen are capable of affecting translocation frequency but did not meet our threshold for further consideration because of either incomplete gene suppression (e.g. KU70; Supplementary Fig. 2b) or excessive toxicity from the shRNA (e.g. MRE11; Supplementary Fig. 2c).”

We agree with the Reviewer's comment that identification of 5 'hits' is a consequence of the somewhat arbitrary thresholds that we set. For example, had we relaxed the cutoffs for fold-change or included those genes that scored strongly but only with one shRNA, we would have identified many more 'hits.' In the interest of reducing false positives, we made our thresholds relatively stringent. As a result, we were able to validate each of those 5 'hits' with other reporters in other human cell lineages. However, there are certainly other shRNA assayed in the screen that affected translocation frequency and thus we have included all of the shRNA sequences and the screening data within the supplementary information to facilitate further study by other groups.

Further, several aspects of the screen are not well explained:

- What are the absolute numbers of GFP positive cells identified in the different lines? This is helpful in interpreting the robustness of the fold-changes reported

We thank the Reviewer for this thoughtful consideration of our screen. Indeed, we spent considerable time setting targets for absolute numbers of GFP⁺ cells needed to measure

robust fold-changes. We apologize that this was not clearly delineated. We have amended the new manuscript to include the following:

“At least 200,000 transgene-positive cells were collected for each replicate to ensure that each shRNA in the library was represented, on average, by >200 transgene-positive cells.”

In addition to the screen, the absolute numbers corresponding to the different translocation reporters for validating the screen hits are included below.

Cell line / assay	% transgene positive cells in wild-type cells	Total cells analyzed / replicate	Absolute number of transgene positive cells/replicate
A549 / zinc finger nuclease mediated rearrangements of AAVS1	0.8-1.2%	100,000	800-1200
HeLa / zinc finger nuclease mediated rearrangements of AAVS1	0.5-1.0%	100,000	500-1000
293T / CRISPR mediated rearrangements of CD4 and CD71	0.1-0.4%	100,000	100-400

- Is there any 'background' GFP positivity even in the absence of the zinc finger nuclease expression.

As a control for ‘background GFP positivity,’ we routinely ran a sample that was infected with only one of the components of the zinc finger nuclease obligate heterodimer (ZFN1 alone) or transfected with only one of the CRISPR/CAS9 gRNAs (targeting *CD4* only). The background was always >30-fold lower for the ZFN-based assay and >10-fold lower for the CRISPR/Cas9-based assay (see data below). The higher background positivity in the CRISPR/CAS9 assay may result from the extra step of antibody staining for CD4, which results in a low level of non-specific positivity.

GFP⁺ background in zinc finger nuclease-mediated translocations

CD4⁺ background in CRISPR/CAS9-mediated translocations

- Do the authors know that every GFP positive event corresponds to a translocation to the AAVS1 locus? Since the GFP or CD4 construct is randomly integrated into the genome it is possible that the upstream cut could also result in deletions that in some circumstances could bring the reporter under the influence of another promoter close to the integration site

This is an excellent point and one that we considered when we performed the screen using two separate reporters. To further address this issue, we performed PCR for the rearrangement between the *AAVS1* locus and the integrated cassette on serial dilutions of cells expressing the AAVS1 zinc finger nucleases. We verified the identity of the PCR product by Sanger sequencing. We were able to extrapolate from the number of wells that were positive for the rearrangement PCR in our serial dilutions that the frequency of transgene positivity measured in our flow cytometry assay (0.8-1.2%) represented a modest over-estimation (~130%) of the frequency of rearrangements recorded by PCR. Similarly, for the CRISPR/CAS9 rearrangements between the *CD4* and *CD71* loci, we were concerned that off-target cutting might result in CD4 expression without chromosomal rearrangement. Therefore, we performed droplet digital PCR (ddPCR) for the CD4/*CD71* rearrangement and compared the resulting ddPCR frequencies with the flow cytometry measurements. We found that CRISPR/CAS9-mediated rearrangements between the *CD4* and *CD71* loci give indistinguishable frequencies by flow cytometry for CD4 expression of ddPCR of fusion transcript (see right).

Comparison between flow cytometry and ddPCR rearrangement measurements

- For each line, do the authors know how many copies of the GFP or CD4 reporters are integrated?

To address this question, we attempted to perform fluorescence *in situ* hybridization (FISH) on the reporter to determine the number of integrations. However, this failed, most likely due to the relatively short length of non-human sequence in the reporter. To reduce the possibility that multiple integrations of the reporter in one of the lines would lead to artifactual results, we opted to use multiple lines for the screen and then validate in still other lines. Finally, we developed the CRISPR/CAS9-mediated rearrangement assay (CRITR, **Figure 3h-o**) to measure rearrangements between endogenous loci in unmodified cells, which technically measures potential events between both *CD4* alleles and both *CD71* alleles.

- Do the three GFP and four CD4-reporter repeats represent different integrations or just repeats on the same integration? This is important as the context of the integration could affect translocation frequency and / or the dependence on particular proteins. It seems that the correlation between the GFP and the CD4 reporters is actually quite poor (Figure 1E). r looks to be low on these plots.

We apologize that this aspect of the screen was not made clear. The three GFP replicates were performed using a single line and the four CD4 replicates were performed using a different, single line. We fully agree with the Reviewer's concern that the 'context of the integration and / or the dependence on particular proteins' could affect translocation frequency. This is precisely why we chose to: 1) use 2 different lines for the screen, 2) validate in a separate line (HeLa) with a separate integration, 3) confirm using different endonucleases (CRISPR/Cas9) in yet another line, and 4) set relatively stringent thresholds in our data analysis (see earlier comment).

2. Why is no Western blot show for PARP3 in Figure 2?

We thank the Reviewer for alerting us to this omission. **Figure 2** has been modified to include a PARP3 western blot.

3. Figure 3 presents a lot of information derived from different cell lines, but [*sic*]

Unfortunately, we are unable to respond to this incomplete comment.

4. The hypersensitivity of PARP3ko cells to pyridostatin is interesting. However, this key result needs to be made more robust and convincing. Is the effect complemented by ectopic expression of PARP3 and is it seen with other G4 stabilising ligands, for instance NMM or PhenDC3, both of which are commercially available?

We thank the Reviewer for these useful experimental suggestions. These additional experiments have been included in the modified manuscript in **Supplementary Figure 4h-j**. The new data show that *PARP3*^{-/-} cells are sensitive to PhenDC3 as well as

pyridostatin and that ectopic expression of PARP3 complements the observed sensitivity. Indeed, we agree that these findings render this key result ‘more robust and convincing.’

5. The experiments in which various components of the DSB response are ChIP'd from around the break site in the CD4 locus are interpreted as showing an influence of G4s in PARP3-deficient cells. However, this is an over interpretation of a correlation. BLM occupancy, for instance (Figure 4h & j) is increased around the break. However, this does not necessarily have anything to do with G4s. BLM does indeed play a role in processing G4s, but it also plays a role in double strand break repair through Holliday junction dissolution, among other functions. In both the case of CD4 and ESR1, BLM occupancy is associated with a break and may not be related to the presence of G4s. For instance, is its recruitment modulated by exposure to pyridostatin?

We thank the Reviewer for this creative experimental suggestion. We performed ChIPs for both RPA and BLM at the targeted break at the CD4 locus with the addition of pyridostatin treatment (see below). We observed that pyridostatin treatment led to a decrease in RPA recruitment around the targeted DSB (left panel), consistent with the notion that stabilization of G4 DNA in the vicinity of a DSB prevents RPA binding.

However, we also observed a decrease in BLM binding around the targeted break (right panel) in contrast to our prediction that stabilization of G4 DNA in the vicinity of the break would increase BLM binding. Upon further consideration, we noted that

BLM chromatin staining following G4 DNA stabilization

pyridostatin treatment results in a strong and global stabilization of G4 DNA throughout the genome. As G4 DNA is thought to be a physiological substrate for BLM helicase, this global increase in G4 DNA could lead to more BLM on chromatin overall but less BLM at any given lesion (*i.e.*, BLM protein is limiting and global G4 stabilization results in dilution of the protein across the genome). To address this, we assayed BLM chromatin association globally by performing immunofluorescence for BLM on CSK-extracted nuclei. Upon treatment with pyridostatin, we observed that overall BLM staining increases (see left). This global increase on chromatin could explain the local decrease that we observe at a targeted DSB making these ChIP results difficult to interpret. We hope the Reviewer will agree that further

exploration of the recruitment of BLM in the context of global G4 DNA stabilization is out-of-scope for this single manuscript, which focuses primarily on the translocation screen and PARP3.

6. The G4 antibody immunoprecipitations are also incomplete. The pyridostatin-induced enrichment at the telomere and possibly at the MYC locus is clear. However, Figure 4k does not include the CD4 locus, which would be expected to give a significantly greater signal than ESR1. Does it?

Following the Reviewer's thoughtful suggestion, we examined the G4 DNA signal captured by hf2 at the undamaged *CD4* locus. In these experiments, we observed a modest but statistically significant enrichment in the hf2 signal with pyridostatin at *CD4* (see left, $p=0.0125$). Of note, the extent of G4 DNA is likely to be enhanced by a DSB, which would expose single-stranded DNA and reduce nucleosomal density during DSB processing. Thus, the lack of robust enrichment of G4 DNA at *CD4* in the absence of a targeted break is not evidence against the enrichment after a DSB.

Likewise, it would be predicted that if the experiment shown in Figure 4i was repeated at the ESR1 locus that there would not be enrichment of the G4 antibody signal after inducing a DSB. The proposed model would also predict that inhibition of PARP3 (using ME0328, as show in Figure 4f) would have the same effect, while PDS would increase the effect.

As the Reviewer suggested, we predicted that G4 DNA signal at this locus with very little predicted G4 DNA structure would not be increased by PARP3 deficiency. Results of this experiment are included in **Supplementary Figure 10a** and excerpted below along with the results from the *CD4* locus for comparison. We found that while a DSB at the *CD4* locus resulted in significantly enhanced G4 DNA in *PARP3*^{-/-} cells (but not wild-type), DSB at the *ESR1* locus did not lead to an increase in the G4 DNA signal regardless of the genotype.

The Reviewer asked whether ‘PDS would increase the [hf2] effect.’ We regret that the experimental conditions were not clear from the text. All of the samples in Figure 4I were treated with pyridostatin (PDS). The corresponding figure legend is excerpted below:

“(I) Quantification of G4 DNA content in the indicated genotypes at the *CD4* locus measured by IP with hf2 antibody 24 hours after pyridostatin (PDS) treatment and 18 hours after transfection with CRISPR/Cas9 with or without gRNA for *CD4*.”

To emphasize this point, we have added language to the Results section of the manuscript and added “PDS” treatment to the Figure as well. In optimizing these experiments, we observed that a G4 DNA signal was not reliably observable by IP unless the cells were first treated with pyridostatin to stabilize the secondary structures.

7. In Figure 5, the quantitation of foci is highly problematic. Given the picture of gammaH2Ax foci show in 5a, the precision and error shown in the quantitation in Figure 5b seems remarkable. Even so, there is only a significant difference at one time point. The G4 antibody foci are even more worrying as no examples of the IF pictures are shown, and again there is basically no significant effect overall.

We apologize that the method used to quantitate foci was unclear and that we did not provide adequate visual representations of the IF images in **Figure 5**. To remedy these points, we have added a specific description of our foci quantification in the updated Methods (excerpted below):

“ImageJ FIJI²¹ was used to quantitate foci in all immunofluorescence experiments. Images from the vehicle-treated condition were used to set an appropriate noise tolerance for each antibody stain. The same noise threshold was used for all images in a given experiment (noise tolerances were as follows: 10 for γ H2AX (JBW301), 10 for G4 DNA (1H6), and 5 for 53BP1 (Bethyl)). The ‘Find Maxima’ tool was used to count foci for the indicated number of nuclei.”

In addition, we have added complete panels of the IF experiments in new Figures (**Supplementary Figures 11-13**), including a complete panel of the G4 foci (**Supplementary Figure 13**) as suggested by the Reviewer.

8. The conclusion that 'resection is suppressed' is not supported by the data presented in Figure 6. The ChIP experiments show only patchy significance and, I'm afraid, are not convincing.

We thank the Reviewer for these comments. We have edited the language in the manuscript to reflect a more conservative interpretation of our ChIP and IF findings with respect to RPA (excerpted below):

“PARP3 promotes CtIP and RPA deposition at DNA DSBs

We hypothesized that the increased prevalence of G4 structures in PARP3-deficient cells would suppress binding by the endonuclease CtIP which converts DSB ends into single-stranded DNA tails that are intermediates for HR. In fact, the efficiency of HR at a single DSB was reduced in PARP3-depleted cells containing the DR-GFP reporter (**Supplementary Fig. 10f-h**), consistent with a previous report²². Cutting by CAS9 at the *CD4* locus was similar in wild-type and *PARP3*^{-/-} cells (**Figure 6a**), but *PARP3*^{-/-} cells had less CtIP accumulation flanking the DSB (**Figure 6b**). The ChIP signal given by the CtIP antibody was validated by siRNA specificity control shown in **Supplementary Fig. 9c**. Processing by CtIP facilitates deposition of the single-stranded DNA binding protein RPA. *PARP3*^{-/-} cells also had reduced RPA deposition around the targeted DSB (**Figure 6c**) but significantly increased γ H2AX signal (**Figure 6d**). The latter finding is consistent with a report that depletion of PARP3 leads to persistence of γ H2AX foci²³. In contrast, we did not observe any significant PARP3-dependent differences in CtIP or RPA deposition or γ H2AX signal at the targeted DSB in *ESR1* (**Supplementary Fig. 14a-c**), where no G4 DNA is expected to form. Furthermore, we examined RPA occupancy following targeted DSB at six additional loci: three in G4-rich regions and three in G4-poor regions (**Supplementary Fig. 15**). We observed that RPA deposition was significantly reduced in *PARP3*^{-/-} cells (compared to wild-type cells) near DSBs in G4-rich regions (**Supplementary Fig. 15a-c**) but not near DSBs in G4-poor regions (**Supplementary Fig. 15d-e**). Together, these data suggest that at DSBs in G4-rich regions, PARP3-deficiency inhibits CtIP and RPA deposition, leading to a delay in repair of these breaks signaled by increased γ H2AX.”

With regard to the ‘patchy significance’ of the RPA ChIP data, we concur that the differences are modest, yet our analysis reveals that they are significant (p -values < 0.05). Furthermore, the findings regarding RPA in the ChIP data at a targeted DSB are corroborated by the IF data using RPA to stain foci following damage by ionizing radiation. Importantly, we have included a statement about the relatively small absolute differences and the need for further validation in additional systems within the Discussion.

Reviewer #3 (Remarks to the Author):

In the manuscript “PARP3 is a promotor of chromosomal rearrangement and limits G4 DNA“ by Day et al. the authors identified two factors, i.e. UBC9 and RAD50, as suppressors of chromosomal rearrangement, and three factors, i.e., 53BP1, DDB1, and PARP3, as promotors of chromosomal rearrangement in two different human cell lines. In the following they studied the role of PARP3 in these processes in greater detail. They convincingly demonstrate that PARP3 is associated with the regulation of G4 DNA in response to DNA damage. Specifically, they showed that PARP3 KO cells are sensitive to pyridostatin-induced DNA double strand breaks. Furthermore, the authors show that PARP3 supports the processing of DSBs into single-stranded DNA and cooperates with BLM helicase in NHEJ repair.

The study is comprehensive, for the most part the data is of high quality, the methods section is detailed, and the manuscript is well written. The conclusions drawn by the

authors and the model developed are supported by their results. In principal, I support publication of this manuscript after considering the following issues:

Fig 2: Could the authors indicate, why they marked some siRNAs by a star? The linear regression analyses should be extended by a statement on the P values.

We apologize for the omissions. The shRNA marked by asterisks in **Figure 2** are those that scored in the original screen. The figure legend has been corrected to include this information. In addition, we have added the *p*-values for the linear regression analyses to the Figure legend.

Fig. 3: Mislabeling of panel b. Scale bars are missing in microscopic pictures of panel j. Size of images in panel j should be increased and image quality should be improved. Unit is missing at the x-axis of panel p.

Again, we are grateful to the Reviewer for finding these errors. In the resubmitted manuscript, the duplicate panels of Figure 3 have been corrected, the size and quality of panel j have been increased and a scale bar has been added, and the correct units have been indicated on the x axis of panel p.

Fig. 4: Panel e: Since a genetic knock-out cannot directly be compared to siRNA-mediated knock-down, PARP3 knock-down should be included in the experiment shown in panel e. Furthermore, the PARP2 Western blot is not convincing. There seems to be a prominent band cut right at the lower edge. In panels h, j, and l, units at the x-axis are missing.

Based on the Reviewer's comment, we repeated the experiment in **Figure 4e** (colony formation to assay sensitivity to pyridostatin) to enable comparison with siPARP1 and siPARP2. These new results show that siPARP3 confers sensitivity to pyridostatin treatment (included in **Figure 4e** and **Supplementary Figure 4k**). In addition, we repeated the PARP2 western blot with a different antibody and have included the new, cleaner blot in **Figure 4e**. Finally, we corrected the x axis in panels **4h, j, and l** (and in all of the other ChIP experiments as well) to include proper units.

Fig. 5: Evaluation of 53BP1 foci and their degree of colocalization with γ H2A.X foci should be analyzed. Scale bars missing in panel a. Quality of images in panel a is not satisfactory. Color code between panels b, c, and d is inconsistent. Choose different color for wild-type (?).

We thank the Reviewer for these suggestions. We repeated the IF experiments in **Figure 5a-c** and included 53BP1. These new results along with analysis of degree of colocalization between 53BP1 and γ H2AX can be found in **Supplementary Figure 10b-e** (and left). We used the FIJI plug-in Jacop to calculate degree of colocalization.²⁴ We are grateful to the Reviewer for catching these inconsistencies. We have added scale bars, improved the image quality, and included a complete IF

time course in new **Supplementary Figures 11-13**. In addition, we have corrected **Figure 5a-d** to use a unified color code here and in the supplementary figures as well.

Can the authors explain the decrease in G4 DNA after 6 hours in “PARP3^{-/-}, ad b-gal” cells. In panel f, it is not clear which groups have been compared for statistical analysis. 2-Way ANOVA testing seems to be the appropriate statistical test here.

We were similarly puzzled by the decrease in G4 DNA after 6 hours represented in **Figure 5c**. We considered the following possible explanations: 1) BLM or another helicase that unwinds DNA secondary structures (such as Werner’s helicase) might unwind the G4 DNA or 2) changes in chromatin structure or loading of repair proteins could block antibody binding giving a falsely reduced readout. Although it is outside the scope of this manuscript, the question of kinetics at the G4 DNA break is an area we are actively investigating. We apologize for the confusion regarding the statistical analysis in **Figure 5e**. We have amended the figure to clearly indicate which groups are being compared. In addition, we have performed a 2-way ANOVA, reported in the legend, as suggested by the Reviewer, to offer an additional statistical analysis of these data:

“A two-way ANOVA (alpha set at 0.05) gave the following *p*-values for cell cycle phases: G1, *p* = 0.5955, S, *p* = 0.0152, and G2, *p* = 0.0111.”

Fig. 6: Units at x-axis are missing in panels b,c,and d. Scale bars are missing in panel e. Can the image quality in panel e be increased?

We have corrected the labeling of the x axes in b, c, and d to include appropriate units. We have added scale bars to Figure 6e and increased the image quality.

Fig. 7. In panels a and b the x-axes could be presented as continuous data (instead of categorical data). The extent of colocalization of 53BP1 and γ -H2A.X foci should be analyzed. Panel d could be presented as a separate Figure.

We thank the Reviewer for these suggestions. We have re-plotted the x-axes as continuous data. In addition, we analyzed the degree of colocalization between γ H2AX and 53BP1 and included this analysis in **Supplementary Figure 16e**. Finally, panel d has been moved to Figure 8.

References

1. Rodriguez, R. et al. Small-molecule-induced DNA damage identifies alternative DNA structures in human genes. *Nat Chem Biol* **8**, 301-10 (2012).
2. Chambers, V.S. et al. High-throughput sequencing of DNA G-quadruplex structures in the human genome. *Nat Biotechnol* **33**, 877-81 (2015).
3. Huang da, W., Sherman, B.T. & Lempicki, R.A. Bioinformatics enrichment tools: paths toward the comprehensive functional analysis of large gene lists. *Nucleic Acids Res* **37**, 1-13 (2009).
4. Huang da, W., Sherman, B.T. & Lempicki, R.A. Systematic and integrative analysis of large gene lists using DAVID bioinformatics resources. *Nat Protoc* **4**, 44-57 (2009).
5. Wu, W.Q., Hou, X.M., Li, M., Dou, S.X. & Xi, X.G. BLM unfolds G-quadruplexes in different structural environments through different mechanisms. *Nucleic Acids Res* **43**, 4614-26 (2015).
6. Chatterjee, S. et al. Mechanistic insight into the interaction of BLM helicase with intra-strand G-quadruplex structures. *Nat Commun* **5**, 5556 (2014).
7. Sun, H., Karow, J.K., Hickson, I.D. & Maizels, N. The Bloom's syndrome helicase unwinds G4 DNA. *J Biol Chem* **273**, 27587-92 (1998).
8. Huber, M.D., Lee, D.C. & Maizels, N. G4 DNA unwinding by BLM and Sgs1p: substrate specificity and substrate-specific inhibition. *Nucleic Acids Res* **30**, 3954-61 (2002).
9. Nguyen, G.H. et al. Regulation of gene expression by the BLM helicase correlates with the presence of G-quadruplex DNA motifs. *Proc Natl Acad Sci USA* **111**, 9905-10 (2014).
10. Johnson, J.E., Cao, K., Ryvkin, P., Wang, L.S. & Johnson, F.B. Altered gene expression in the Werner and Bloom syndromes is associated with sequences having G-quadruplex forming potential. *Nucleic Acids Res* **38**, 1114-22 (2010).
11. Drosopoulos, W.C., Kosiyatrakul, S.T. & Schildkraut, C.L. BLM helicase facilitates telomere replication during leading strand synthesis of telomeres. *J Cell Biol* **210**, 191-208 (2015).
12. Sarkies, P. et al. FANCD1 coordinates two pathways that maintain epigenetic stability at G-quadruplex DNA. *Nucleic Acids Res* **40**, 1485-98 (2012).

13. Lam, E.Y., Beraldi, D., Tannahill, D. & Balasubramanian, S. G-quadruplex structures are stable and detectable in human genomic DNA. *Nat Commun* **4**, 1796 (2013).
14. Henderson, A. et al. Detection of G-quadruplex DNA in mammalian cells. *Nucleic Acids Res* **42**, 860-9 (2014).
15. Hoffmann, R.F. et al. Guanine quadruplex structures localize to heterochromatin. *Nucleic Acids Res* **44**, 152-63 (2016).
16. Carle, C.M., Zaher, H.S. & Chalker, D.L. A Parallel G Quadruplex-Binding Protein Regulates the Boundaries of DNA Elimination Events of *Tetrahymena thermophila*. *PLoS Genet* **12**, e1005842 (2016).
17. Hansel-Hertsch, R. et al. G-quadruplex structures mark human regulatory chromatin. *Nat Genet* **48**, 1267-72 (2016).
18. Leroy, B. et al. Analysis of TP53 mutation status in human cancer cell lines: a reassessment. *Hum Mutat* **35**, 756-65 (2014).
19. Yusein-Myashkova, S., Stoykov, I., Gospodinov, A., Ugrinova, I. & Pasheva, E. The repair capacity of lung cancer cell lines A549 and H1299 depends on HMGB1 expression level and the p53 status. *J Biochem* **160**, 37-47 (2016).
20. Lin, Y.C. et al. Genome dynamics of the human embryonic kidney 293 lineage in response to cell biology manipulations. *Nat Commun* **5**, 4767 (2014).
21. Schindelin, J. et al. Fiji: an open-source platform for biological-image analysis. *Nat Methods* **9**, 676-82 (2012).
22. Beck, C. et al. PARP3 affects the relative contribution of homologous recombination and nonhomologous end-joining pathways. *Nucleic Acids Res* (2014).
23. Rulten, S.L. et al. PARP-3 and APLF function together to accelerate nonhomologous end-joining. *Mol Cell* **41**, 33-45 (2011).
24. Bolte, S. & Cordelieres, F.P. A guided tour into subcellular colocalization analysis in light microscopy. *J Microsc* **224**, 213-32 (2006).

Reviewers' Comments:

Reviewer #1 (Remarks to the Author)

The revised manuscript benefits by inclusion of the RNA-seq analysis supporting the view that PARP3 does not regulate transcription of repair genes. The manuscript benefits by inclusion of previously missing controls, pointed out by all reviewers. However, the revised manuscript remains unsuccessful at connecting PARP3 and G4 DNA. Attempts to make this connection are based on unfounded assumptions about specificity of BLM for G4 DNA, as is made very clear in the rebuttal.

The authors wish to argue that BLM accumulation near breakpoints that participate in translocation provides evidence for quadruplex structures at those breakpoints. The rebuttal states that the revised version of the manuscript now rationalizes this argument as follows:

"The BLM helicase is known to bind and unwind G4 DNA *in vitro*⁵⁻⁸ and can serve as a surrogate to map G4 DNA locations *in vivo*^{9,10}. In addition, BLM facilitates replication through G4 DNA in telomeric sequences *in vivo*¹¹ and BLM-deficient cells exhibit G4 DNA motif-associated epigenetic instability¹².

While BLM does bind to and unwind quadruplexes *in vivo*, it has many other functions which could explain its accumulation at sites of damage, as detailed by Reviewer 2. BLM cannot serve as a surrogate to map G-quadruplexes *in vivo*, nor do the references cited (Ref. 9 and 10) make that claim or demonstrate that to be the case. Genomewide ChIP-Seq analysis would be necessary to establish that BLM can serve as a "surrogate" to map G4 DNA locations *in vivo*; while the references cited report on the role of BLM in transcriptional regulation, identifying correlations between G4 motifs and transcriptional regulation by BLM. Moreover, epigenetic instability at G4 motifs reported in Ref. 12 was observed in the absence of two G4 helicases, FANCD1 and BLM, in mutant derivatives of chicken DT40 cells; and is currently viewed as reflecting replication stress (see Svikovic and Sale 2016 PMID 27876548).

A limited understanding of whether and how G-quadruplexes contribute to genomic instability in the human genome is also evident in the use of pyridostatin. The results of Rodriguez et al. do not support the notion that pyridostatin causes damage only at quadruplexes or at all quadruplexes, though the manuscript at hand treats this reagent as though this were the case.

Separate from the involvement of G-quadruplexes, the mechanism by which PARP3 promotes translocation could be of interest. However, the results reported are relatively modest in magnitude, and few target sites/genes/cell types are examined, and it would require considerable systematic analysis to know if these results are robust or would prove general. For example, the net 2.5-fold accumulation of BLM at targeted DSBs is a relatively modest effect, especially in the absence of stronger controls for specificity of the anti-BLM antibodies. The previous review requested this, and the most appropriate control would have been a western blot showing that IP with the anti-BLM antibody brought down a single band of the predicted size, a control that typically accompanies ChIP-Seq analysis. The control added (Supp. Fig. 9b) shows that the ChIP signal assigned to BLM is reduced in cells treated with siBLM. The reduction of the BLM signal is shown only at CD4, and reduction is only 2.5-fold. Is this background? If so, then why is it higher than the EV control? Why the difference with the CtIP signal? Is one time point sufficient to draw any conclusion?

To take another example, the authors propose that RPA does not accumulate at DSBs at "G4-rich" sites in the absence of PARP3 (Supp. Fig. 15). But this is only a 2- to 3-fold effect, which is

modest, and only shown for six different sites. Obvious confounders are unequal efficiency of targeting of different genomic loci by CRISPR/Cas9 (possibly consistent with the unexplained differences in magnitude of RPA occupancy at the different sites tested); kinetic differences in repair among sites; as well as features that may differ among the targets including transcription, transcription direction, chromatin structure, replication, replication direction....

Reviewer #2 (Remarks to the Author)

The authors have worked hard to very carefully address the comments of all the reviewers, including my own. I still have reservations concerning the limitations of using G4 ligands and the anti-G4 antibodies to infer the the role of G4s in explaining the effects of PARP3 in promoting HR and translocations. However, given these caveats associated with the experimental approach, I think that the authors have gone as far as is it reasonable to expect currently in providing support for their hypothesis. It will be for future studies to test the idea further with both biochemical and more refined genetic approaches, and I hope this paper will stimulate such work.

Reviewer #3 (Remarks to the Author)

All my points have been addressed satisfactorily. I recommend publication of the manuscript as it is.

NCOMMS-16-18904B Rebuttal
January 11, 2017

Reviewers' comments:

Reviewer #1 (Remarks to the Author):

The revised manuscript benefits by inclusion of the RNA-seq analysis supporting the view that PARP3 does not regulate transcription of repair genes. The manuscript benefits by inclusion of previously missing controls, pointed out by all reviewers.

We thank the Reviewer for making these suggestions that significantly improved the manuscript.

However, the revised manuscript remains unsuccessful at connecting PARP3 and G4 DNA. Attempts to make this connection are based on unfounded assumptions about specificity of BLM for G4 DNA, as is made very clear in the rebuttal.

The authors wish to argue that BLM accumulation near breakpoints that participate in translocation provides evidence for quadruplex structures at those breakpoints. The rebuttal states that the revised version of the manuscript now rationalizes this argument as follows:

“The BLM helicase is known to bind and unwind G4 DNA in vitro⁵⁻⁸ and can serve as a surrogate to map G4 DNA locations in vivo^{9,10}. In addition, BLM facilitates replication through G4 DNA in telomeric sequences in vivo¹¹ and BLM-deficient cells exhibit G4 DNA motif-associated epigenetic instability¹².”

While BLM does bind to and unwind quadruplexes in vivo, it has many other functions which could explain its accumulation at sites of damage, as detailed by Reviewer 2. BLM cannot serve as a surrogate to map G-quadruplexes in vivo, nor do the references cited (Ref. 9 and 10) make that claim or demonstrate that to be the case. Genomewide ChIP-Seq analysis would be necessary to establish that BLM can serve as a “surrogate” to map G4 DNA locations in vivo; while the references cited report on the role of BLM in transcriptional regulation, identifying correlations between G4 motifs and transcriptional regulation by BLM. Moreover, epigenetic instability at G4 motifs reported in Ref. 12 was observed in the absence of two G4 helicases, FANCD1 and BLM, in mutant derivatives of chicken DT40 cells; and is currently viewed as reflecting replication stress (see Svikovic and Sale 2016 PMID 27876548).

The Reviewer is correct. To our embarrassment, the previous version of the manuscript contained over-interpretations of data in the literature relevant to BLM. We have removed several statements regarding BLM from the manuscript, most notably that BLM might serve as a ‘surrogate’ to identify G4-quadruplexes. Instead, we state that the role of BLM that accumulates at DNA DSBs in the absence of PARP3 remains unclear and requires further investigation. We suggest that unwinding G4 DNA is only one possible

explanation among many but is concordant with *in vitro* data regarding the activity of BLM.

The section of the manuscript in question now reads,

“The BLM helicase is known to bind and unwind G4 DNA *in vitro*¹⁻⁴. In addition, BLM facilitates replication through G4 DNA in telomeric sequences *in vivo*⁵. However, the role of BLM in promoting DSB repair specifically within regions that harbor abundant G4 DNA remains unclear. Based on the *in vitro* data, one possibility is that BLM facilitates the repair of DSBs within G4-rich regions by unwinding secondary DNA structures.”

“After induction of a DSB at the *CD4* locus, BLM recruitment was significantly increased in *PARP3*^{-/-} cells by ChIP relative to wild-type cells (**Figure 4h**). One possible explanation for this finding is that BLM is recruited to unwind G4 DNA that is more abundant in *PARP3*^{-/-} cells.”

In addition, we have included an expanded section in the Discussion and modified the model in **Figure 8** to emphasize that the role of BLM at DSBs in G4-rich regions remains unclear; that BLM unwinds secondary DNA structures is only one possibility. The text in the Discussion reads:

“The role of BLM helicase at DNA DSBs within G4-rich regions remains unclear. We have shown that BLM accumulates to a greater extent in the absence of PARP3 at a DNA DSB targeted to a region of abundant G4 DNA potential. Given that BLM has been shown to unwind G4 DNA *in vitro*³, one possibility is that BLM unwinds G4 DNA in the context of DSBs *in vivo* as well. However, several other functions for BLM have been reported during repair of DNA DSBs including unwinding Holliday junctions⁶, promoting DNA DSB resection by recruiting EXO1 and DNA2⁷, and blocking recombination by disrupting RAD51 nucleoprotein filaments^{8,9}. Further experiments are required to distinguish between these mechanistic possibilities.”

A limited understanding of whether and how G-quadruplexes contribute to genomic instability in the human genome is also evident in the use of pyridostatin. The results of Rodriguez et al. do not support the notion that pyridostatin causes damage only at quadruplexes or at all quadruplexes, though the manuscript at hand treats this reagent as though this were the case.

We share the Reviewer’s concern that G4 stabilizing ligands such as pyridostatin are imperfect tools to study G-quadruplexes. To improve the manuscript, and at the Reviewers’ previous suggestion, we added corroborative data using PhenDC3, a second G4 stabilizer. To further convey the concern over these reagents, we have added additional language throughout the manuscript. In the Results section:

“In fact, pyridostatin leads to cell cycle arrest by inducing replication- and transcription-dependent DNA damage that preferentially, but not exclusively, occurs

within gene bodies enriched for G-quadruplex forming sequences¹⁰. Although pyridostatin is an imperfect tool for studying G4 DNA, two recent studies have reported clear enrichments for G4 motifs at sites of pyridostatin-induced damage¹⁰ and pyridostatin binding¹¹. Therefore, we used pyridostatin as a tool to preferentially target regions that are enriched for G4 DNA. Furthermore, the observation that *PARP3*^{-/-} cells are also sensitive to PhenDC3, a structurally distinct G4 DNA ligand, lends support to the hypothesis that *PARP3*^{-/-} cells are preferentially susceptible to stabilization of G4 DNA.”

And in the Discussion:

“A key insight came from the finding that PARP3-deficient cells are highly sensitive to pyridostatin and PhenDC3, small molecules that preferentially but not exclusively stabilize and damage regions of G4 DNA. This raised the possibility that PARP3-deficient cells are susceptible to G4 DNA-mediated damage. In fact, *PARP3*^{-/-} cells had increased G4 DNA compared to wild-type cells at sites flanking DSBs within G4-rich regions. Taken together, these findings suggest that *PARP3*^{-/-} cells accumulate more G4 DNA than wild-type cells.”

“There are several aspects of the model that require experimental clarification: 1) As discussed above, the specificity of G4 DNA ligands is imperfect¹⁰. Therefore, it will be important to conclusively define the mechanism of sensitization to pyridostatin and PhenDC3 in the absence of PARP3. The RNA-seq experiments described in **Supplementary Figures 5-8** represent an important first step but further experiments are needed.”

Separate from the involvement of G-quadruplexes, the mechanism by which PARP3 promotes translocation could be of interest. However, the results reported are relatively modest in magnitude, and few target sites/genes/cell types are examined, and it would require considerable systematic analysis to know if these results are robust or would prove general.

Among the recent publications describing genetic factors involved in translocations in mammalian cells, most report the involvement of $\leq 1-2$ genes at $\leq 1-2$ pairs of loci in $\leq 1-2$ different cell types¹²⁻¹⁶. The only exceptions are Ghezraoui et al., 2014, which profiled 3 pairs of loci in 3 cell types¹⁷, and Bhargava *et al.*, 2017, which implicated 5 genes but only at 1 locus pair in 1 cell type¹⁸. Our study exceeds these publications by screening 169 genes and describing 5 genes, 4 pairs of loci, and 3 different cell types. We agree completely with Reviewer #1 that a “systematic analysis” is greatly needed across the genome, but considering the available technology, we believe our manuscript represents a highly significant advance beyond the previous literature.

For example, the net 2.5-fold accumulation of BLM at targeted DSBs is a relatively modest effect, especially in the absence of stronger controls for specificity of the anti-BLM antibodies. The previous review requested this, and the most appropriate control would have been a western blot showing that IP with the anti-BLM antibody brought

down a single band of the predicted size, a control that typically accompanies ChIP-Seq analysis.

The control added (Supp. Fig. 9b) shows that the ChIP signal assigned to BLM is reduced in cells treated with siBLM. The reduction of the BLM signal is shown only at CD4, and reduction is only 2.5-fold. Is this background? If so, then why is it higher than the EV control? Why the difference with the CtIP signal? Is one time point sufficient to draw any conclusion?

These controls as described by Reviewer #1 in the latest round of comments are now included in **Supplementary Figure** panels **9b and 9d** of the revised submission. We explain that the IP using our antibody resulted in a single band consistent with BLM but also a single smaller band. The latter could represent an off-target binder, a BLM degradation product or simply an experimental artifact. Thus, we have also included the siBLM data, which confirms that knockdown of BLM reduces ChIP signal to background (i.e., not statistically significant compared to EV control), strongly arguing for specificity to BLM. It is unclear to us how a time course experiment would further strengthen this data supporting specificity of the antibody.

To take another example, the authors propose that RPA does not accumulate at DSBs at “G4-rich” sites in the absence of PARP3 (Supp. Fig. 15). But this is only a 2- to 3-fold effect, which is modest, and only shown for six different sites. Obvious confounders are unequal efficiency of targeting of different genomic loci by CRISPR/Cas9 (possibly consistent with the unexplained differences in magnitude of RPA occupancy at the different sites tested); kinetic differences in repair among sites; as well as features that may differ among the targets including transcription, transcription direction, chromatin structure, replication, replication direction....

We thank the Reviewer for these observations. We agree that our model would be advanced by a genome-wide profiling experiment of a greater number of targeted DNA double-strand breaks such as by ChIP-seq. We feel that this is an ideal direction for future studies. In addition, we agree that there are many possible confounding factors when targeting multiple genomic loci with DNA damage; those mentioned by the Reviewer constitute important caveats to bear in mind when considering these data. However, we also believe that a highly consistent effect observed across 8 sites is very unlikely to be by chance. Nonetheless, we have expanded the Discussion of the manuscript (excerpted below).

“Our data showing PARP3-dependent differences in accumulation of G4 DNA and BLM helicase at DNA DSBs is based on profiling of two targeted loci: one with high G4 DNA potential (*CD4*) and one lacking the potential to form G4 DNA structures (*ESR1*). Altogether, our data showing PARP3-dependent differences in RPA accumulation assessed 8 loci with varying G4 DNA potential, however, this remains a relatively small set of loci. Therefore, to strengthen our model and overcome potential artifacts resulting from comparison of a small number of loci, data from a larger set of loci with varying G4 DNA content is needed. Techniques that will allow

profiling of the PARP3-dependency of G4 DNA in a genome-wide manner were recently reported¹⁹. The combination of these techniques with a larger set of targeted DNA DSBs will provide important future data to refine or even refute our model.”

Reviewer #2 (Remarks to the Author):

The authors have worked hard to very carefully address the comments of all the reviewers, including my own. I still have reservations concerning the limitations of using G4 ligands and the anti-G4 antibodies to infer the the role of G4s in explaining the effects of PARP3 in promoting HR and translocations. However, given these caveats associated with the experimental approach, I think that the authors have gone as far as is it reasonable to expect currently in providing support for their hypothesis. It will be for future studies to test the idea further with both biochemical and more refined genetic approaches, and I hope this paper will stimulate such work.

We thank Reviewer #2 for this assessment. We have revised language throughout the manuscript to reflect the limitations of using G4 ligands as an experimental tool (see above).

Reviewer #3 (Remarks to the Author):

All my points have been addressed satisfactorily. I recommend publication of the manuscript as it is.

We thank Reviewer #3 for this response.

References

1. Wu, W.Q., Hou, X.M., Li, M., Dou, S.X. & Xi, X.G. BLM unfolds G-quadruplexes in different structural environments through different mechanisms. *Nucleic Acids Res* **43**, 4614-26 (2015).
2. Chatterjee, S. et al. Mechanistic insight into the interaction of BLM helicase with intra-strand G-quadruplex structures. *Nat Commun* **5**, 5556 (2014).
3. Sun, H., Karow, J.K., Hickson, I.D. & Maizels, N. The Bloom's syndrome helicase unwinds G4 DNA. *J Biol Chem* **273**, 27587-92 (1998).
4. Huber, M.D., Lee, D.C. & Maizels, N. G4 DNA unwinding by BLM and Sgs1p: substrate specificity and substrate-specific inhibition. *Nucleic Acids Res* **30**, 3954-61 (2002).
5. Drosopoulos, W.C., Kosiyatrakul, S.T. & Schildkraut, C.L. BLM helicase facilitates telomere replication during leading strand synthesis of telomeres. *J Cell Biol* **210**, 191-208 (2015).
6. Bizard, A.H. & Hickson, I.D. The dissolution of double Holliday junctions. *Cold Spring Harb Perspect Biol* **6**, a016477 (2014).

7. Nimonkar, A.V. et al. BLM-DNA2-RPA-MRN and EXO1-BLM-RPA-MRN constitute two DNA end resection machineries for human DNA break repair. *Genes Dev* **25**, 350-62 (2011).
8. Wu, L., Davies, S.L., Levitt, N.C. & Hickson, I.D. Potential role for the BLM helicase in recombinational repair via a conserved interaction with RAD51. *J Biol Chem* **276**, 19375-81 (2001).
9. Schwendener, S. et al. Physical interaction of RECQ5 helicase with RAD51 facilitates its anti-recombinase activity. *J Biol Chem* **285**, 15739-45 (2010).
10. Rodriguez, R. et al. Small-molecule-induced DNA damage identifies alternative DNA structures in human genes. *Nat Chem Biol* **8**, 301-10 (2012).
11. Chambers, V.S. et al. High-throughput sequencing of DNA G-quadruplex structures in the human genome. *Nat Biotechnol* **33**, 877-81 (2015).
12. Weinstock, D.M., Brunet, E. & Jasin, M. Formation of NHEJ-derived reciprocal chromosomal translocations does not require Ku70. *Nat Cell Biol* **9**, 978-81 (2007).
13. Simsek, D. & Jasin, M. Alternative end-joining is suppressed by the canonical NHEJ component Xrcc4-ligase IV during chromosomal translocation formation. *Nat Struct Mol Biol* **17**, 410-6 (2010).
14. Simsek, D. et al. DNA ligase III promotes alternative nonhomologous end-joining during chromosomal translocation formation. *PLoS Genet* **7**, e1002080 (2011).
15. Zhang, Y. & Jasin, M. An essential role for CtIP in chromosomal translocation formation through an alternative end-joining pathway. *Nat Struct Mol Biol* **18**, 80-4 (2011).
16. Wray, J. et al. PARP1 is required for chromosomal translocations. *Blood* **121**, 4359-65 (2013).
17. Ghezraoui, H. et al. Chromosomal translocations in human cells are generated by canonical nonhomologous end-joining. *Mol Cell* **55**, 829-42 (2014).
18. Bhargava, R., Carson, C.R., Lee, G. & Stark, J.M. Contribution of canonical nonhomologous end joining to chromosomal rearrangements is enhanced by ATM kinase deficiency. *Proc Natl Acad Sci U S A* (2017).
19. Hansel-Hertsch, R. et al. G-quadruplex structures mark human regulatory chromatin. *Nat Genet* **48**, 1267-72 (2016).

Reviewers' Comments:

Reviewer #2 (Remarks to the Author)

The authors have further toned down the interpretation of aspects of their data, particularly those concerning the use of BLM ChIP to detect G4s and the use of the G4 stabilising ligands. This is to be welcomed. This reworking of the ms has gone further to recognise the limitations of the approaches used and the potential disconnect between the experiments performed and their interpretation in terms of the model being advanced. I still think that, despite these limitations, there is much to commend this manuscript.